

# Seasonal influences on surface ozone variability in continental South Africa and implications for air quality

**Tracey Leah Laban[1], Pieter Gideon van Zyl[1*], Johan Paul Beukes[1], Ville Vakkari[2], Kerneels Jaars[1], Nadine Borduas-Dedekind[3], Miroslav Josipovic[1], Anne Mee Thompson[4],**
5    **Markku Kulmala[5], and Lauri Laakso[2]**

1    Unit for Environmental Sciences and Management, North-West University, Potchefstroom, South Africa

2    Finnish Meteorological Institute, Helsinki, Finland

10   3    Department of Environmental Systems Science, ETH Zürich, Zürich, Switzerland

NASA/Goddard Space Flight Center, Greenbelt, Maryland, USA

Department of Physics, University of Helsinki, Finland

*Correspondence to: P.G. van Zyl (pieter.vanzyl@nwu.ac.za)

## Abstract

Although elevated ozone ($O_3$) concentrations are observed in many areas within continental southern Africa, few studies have investigated the regional atmospheric chemistry and dominant
atmospheric processes driving surface $O_3$ formation in this region. The aim of this study was to conduct an assessment of comprehensive continuous surface $O_3$ measurements performed at four sites located in continental South Africa. These sites were representative of regional background (Welgegund and Botsalano) and industrial regions (Marikana and Elandsfontein) in the north-eastern interior in South Africa as indicated by comparison with other sites in this
region. The regional $O_3$ problem was also shown with $O_3$ concentrations being higher than 40 ppb at many sites in the north-eastern interior, while the South African air quality standard limit for $O_3$ was regularly exceeded at the four sites in this study. $O_3$ levels were generally lower at other background sites in the Southern Hemisphere compared to the South African sites, while similar seasonal patterns were observed. The temporal $O_3$ patterns observed at the four sites
resembled typical trends for $O_3$ in continental South Africa, i.e. $O_3$ concentration peaking in late winter and early spring, and daytime $O_3$ peaks associated with increased photochemical production. The seasonal $O_3$ trends observed in continental South Africa were mainly attributed to the seasonal changes in emissions of $O_3$ precursor species and changes in meteorological



conditions. Increased $O_3$ concentrations in winter were indicative of increased emissions of $O_3$ precursors from household combustion for space heating and the concentration of low-level pollutants near the surface. A spring maximum was observed at all the sites, which was attributed to increased regional biomass burning during this time. Source area maps of $O_3$ and

CO indicated significantly higher $O_3$ and CO concentrations associated with air masses passing over a region where a large number of seasonal open biomass burning occurred in southern Africa, which indicated CO associated with open biomass burning as a major source of $O_3$ in continental South Africa. The relationship between $O_3$, $NO_x$ and CO indicated a strong dependence of $O_3$ on CO, while $O_3$ levels remained relatively constant or decreased with

increasing $NO_x$. The seasonal changes in the relationship between $O_3$ and precursors species also reflected the seasonal changes in sources of precursors. The instantaneous production rate of $O_3$, $P(O_3)$, calculated at Welgegund indicated that at least 40% of $O_3$ production occurred in the VOC-limited regime. These relationships between $O_3$ concentrations and $P(O_3)$ with $O_3$ precursor species revealed that large parts of the regional background in continental South

Africa can be considered CO- or VOC-limited, which can be attributed to high anthropogenic emissions of $NO_x$ in the interior of South Africa. It was indicated that the appropriate emission control strategy should be CO (and VOC) reduction associated with household combustion and regional open biomass burning to effectively reduce $O_3$ pollution in continental South Africa.

**Keywords:** ozone ($O_3$) production, $NO_x$-limited, VOC-limited, biomass burning, regional $O_3$, air quality

## 1.    Introduction

Elevated levels of surface (lower troposphere) ozone ($O_3$) have been globally reported for several decades, especially, in North America and Europe, and more recently in Asia, which is generally attributed to increased fossil fuel combustion contributing to increased emissions of $O_3$ precursors (Jaffe and Ray, 2007). High surface $O_3$ concentrations are a serious environmental concern, due to its detrimental impacts on human health, crops and vegetation (National

Research NRC, 1991). Photochemical smog, comprising $O_3$ as a constituent together with other atmospheric oxidants (e.g. nitrogen- and sulphur oxides), is a major air quality concern on an urban and regional scale. Tropospheric $O_3$ is also a greenhouse gas that directly contributes to global warming (IPCC, 2013).



Tropospheric $O_3$ concentrations are regulated by three processes, i.e. chemical production/destruction, atmospheric transport and losses to surface through dry deposition (Monks et al., 2015). The photolysis of $NO_2$ in the presence of sunlight, followed by the addition of the O atom to $O_2$ is the only known way of producing $O_3$ in the troposphere (Logan, 1985):

$$NO_2 + h\nu \longrightarrow O + NO \qquad (1)$$
$$O_2 + O + M \longrightarrow O_3 + M \qquad (2)$$

$O_3$ and nitric oxide (NO) recombine to regenerate $NO_2$, which will once again undergo photolysis to regenerate $O_3$ and NO:

$$O_3 + NO \longrightarrow O_2 + NO_2 \qquad (3)$$

This continuous process is known as the $NO_x$-dependent photostationary state (PSS) and results in no net production or consumption of ozone (null cycle). Net production of $O_3$ ('new ozone') occurs outside the PSS when an atmospheric pool of peroxy radicals ($HO_2$ and $RO_2$) alters the PSS by reacting with NO and producing new $NO_2$ (Cazorla and Brune, 2010). The main source of peroxy radicals is the reaction of the hydroxyl radical ($OH^\bullet$) with volatile organic compounds (VOCs) or carbon monoxide (CO) (Cazorla and Brune, 2010):

$$OH^\bullet + RH + O_2 \longrightarrow H_2O + RO_2 \qquad (4)$$
$$OH^\bullet + CO + O_2 \longrightarrow CO_2 + HO_2 \qquad (5)$$

These organic peroxy radicals or hydroperoxy radicals oxidise atmospheric NO:

$$RO_2 + NO \longrightarrow RO + NO_2 \qquad (6)$$
$$HO_2 + NO \longrightarrow OH + NO_2 \qquad (7)$$

reducing the sink for $O_3$ (Atkinson, 2000), since the resultant $NO_2$ leads to the production of $O_3$ through reaction (1) and (2).

These precursor species can be emitted from natural and anthropogenic sources. Anthropogenic fossil fuel combustion is considered to be the main source of $NO_x$ in South Africa, which include coal-fired power-generation, petrochemical operations, transportation and residential burning (Wells et al., 1996; Held et al., 1996; Held and Mphepya, 2000). Satellite observations indicate a well-known $NO_2$ hotspot over the South African Highveld (Lourens et al., 2012) attributed to industrial activity in the region. CO is produced from three major sources, i.e. fossil fuel combustion, biomass burning, as well as the oxidation of methane ($CH_4$) and VOCs (Novelli et al., 1992). Anthropogenic sources of VOCs are largely due to industrial and vehicular emissions (Jaars et al., 2014), while emissions from vegetation provide the biogenic source (Jaars et al., 2016). Regional biomass burning, which includes household combustion for space heating and cooking, agricultural waste burning and open biomass burning (wild fires), is a





significant source of CO, $NO_x$ and VOCs (Macdonald et al., 2011; Crutzen and Andreae, 1990; Galanter et al., 2000; Simpson et al., 2011) in southern Africa. In addition, stratospheric intrusions of $O_3$-rich air to the free troposphere may also occur that can lead to elevated tropospheric $O_3$ concentrations (Yorks et al., 2009; Lin et al., 2012). The production of $O_3$ from

natural precursor sources, the long-range transport of $O_3$ and the injections from stratospheric $O_3$ contribute to background $O_3$ levels, which is beyond the control of regulators.

Knowledge of the $O_3$ production regime, which is generally classified as either VOC- or $NO_x$-limited, is crucial in designing effective $O_3$ control policies for a given location. However, $O_3$

production has a complex and non-linear dependence on precursor emissions (e.g. National Research NRC, 1991), which makes its atmospheric levels difficult to control (Holloway and Wayne, 2010). The VOC/$NO_x$ ratio has been widely used to categorise an environment as being either $NO_x$- or VOC-limited, since net $O_3$ production requires $NO_x$ and VOCs to exist in specific ratios for the photochemical reaction to occur. $O_3$ formation is $NO_x$-limited when the VOC/$NO_x$

ratio is high, while a low VOC/$NO_x$ ratio indicates that $O_3$ formation is VOC-limited. In the $NO_x$-limited regime, $O_3$ concentrations increase with increasing $NO_x$ and are insensitive to VOCs. Therefore $NO_x$ reductions are most effective in reducing $O_3$ levels. Under VOC-limited conditions, $O_3$ concentrations increase with increasing VOCs and decrease with increasing $NO_x$. VOC reductions will therefore be most effective in reducing $O_3$, while $NO_x$ controls may lead to

$O_3$ increases. There exists a transitional region between the $NO_x$- and VOC-limited regimes where $O_3$ is equally sensitive to each species, and control of both VOC and $NO_x$ might be preferred (National Research NRC, 1991). In general, it is considered that $O_3$ formation in urban areas, close to anthropogenic sources, is VOC-limited, while rural areas distant form source regions are $NO_x$-limited (Sillman, 1999).

Since $O_3$ concentrations are regulated in South Africa, $O_3$ monitoring is carried out across South Africa through a network of air quality monitoring stations established mainly by provincial governments, local municipalities and industries (http://www.saaqis.org.za). High $O_3$ concentrations are observed in many areas within the interior of South Africa that exceed the

South African standard $O_3$ limit (e.g. Laakso et al., 2013), which can be attributed to high anthropogenic emissions of $NO_x$ and VOCs in dense urban and industrial areas (Jaars et al., 2014), regional biomass burning (Lourens et al., 2011) and $O_3$ conducive meteorological conditions (e.g. sunlight). Furthermore, since $O_3$ is a secondary pollutant, high levels of $O_3$ can be found in rural areas downwind of city centres and industrial areas. In order for South Africa to



develop an effective national/provincial management plan to reduce $O_3$ concentrations through controlling $NO_x$ and VOC emissions, it is important to determine whether a region is $NO_x$- or VOC-limited. Results from a photochemical box model study in South Africa, for instance, revealed that the Johannesburg-Pretoria megacity is within a VOC-limited regime (Lourens et
al., 2016).

Previous assessments of tropospheric $O_3$ over continental South Africa focussed on surface $O_3$ (Venter et al., 2012; Laakso et al., 2012; Lourens et al., 2011; Josipovic et al., 2010; Martins et al., 2007; Zunckel et al., 2004), as well as free tropospheric $O_3$ based on soundings and aircraft
observations (Diab et al., 1996; Thompson, 1996; Swap et al., 2003; Diab et al., 2004). Two major field campaigns (SAFARI-92 and SAFARI 2000) were conducted to improve the understanding of the effects of regional biomass burning emissions on $O_3$ over southern Africa. These studies indicated a late winter early spring (August and September) maximum over the region, which was mainly attributed to increased regional biomass burning (wild fires) during this
period. Lourens et al. (2011) also attributed higher $O_3$ concentrations in spring in the Mpumalanga Highveld to increased regional biomass burning. A more recent study demonstrated that $NO_x$ strongly affects $O_3$ levels in the Highveld, especially in winter and spring (Balashov et al., 2014). A regional photochemical modelling study (Zunckel et al., 2006) have attempted to explain surface $O_3$ variability, which found no dominant source/s on elevated $O_3$
levels.

The aim of this study was to provide an up-to-date assessment of the seasonal and diurnal variations in surface $O_3$ concentrations over continental South Africa, as well as to identify local and regional sources of precursors contributing to surface $O_3$. Another significant objective was
to use available ambient data as a means of qualitatively assessing whether $O_3$ formation is $NO_x$- or VOC-limited in different environments. An understanding of the key precursors that control surface $O_3$ production will assist in establishing the $O_3$ production regime, which is critical for the development of an effective $O_3$ control strategy.



## 2. Methodology

### 2.1 Study area and measurement stations

Continuous in-situ $O_3$ measurements obtained from four research stations in the north-eastern interior of South Africa indicated in Fig. 1, which include Botsalano (25.54º S, 25.75º E, 1420 m a.s.l.), Marikana (25.70º S, 27.48º E, 1170 m a.s.l.), Welgegund (26.57 º S, 26.94º E, 1480 m a.s.l.) and Elandsfontein (26.25º S, 29.42º E, 1750 m a.s.l.), were analysed. This region is the largest industrial (indicated by major point sources in Fig. 1) area in South Africa with

substantial gaseous and particulate emissions from numerous industries, domestic fuel burning and vehicles (Lourens et al., 2012; Lourens et al., 2011), while the Johannesburg-Pretoria megacity is also located in this area (Fig. 1). A combination of meteorology and anthropogenic activities has amplified the pollution levels within the region. The seasons in South Africa correspond to typical austral seasons, i.e. winter from June to August, spring from September to

November, summer from December to February and autumn from March to May. The climate is semi-arid with an annual average precipitation of around 400-500 mm (Klopper et al., 2006; Dyson et al., 2015), although there is considerable inter-annual variability associated with El Niño Southern Oscillation (ENSO) phenomena. Precipitation in the north-eastern interior occurs mostly during the austral summer, from October to March, whereas the region is characterised

by a distinct cold and dry season from May to September, i.e. late autumn to mid-spring, during which almost no precipitation occurs. During this period the formation of several inversion layers are present in the region that limit the vertical dilution of air pollution, while more pronounced anticyclonic recirculation of air masses also occur. This synoptic scale meteorological environment leads to an accumulation of pollutants in the lower troposphere in this region,

which can be transported for several days (Tyson and Preston-Whyte, 2000; Garstang et al., 1996). The SAFARI-92 and SAFARI 2000 campaigns indicated that locations in southern Africa thousands of kilometres apart are linked through regional anticyclonic circulation (Swap et al., 2003).



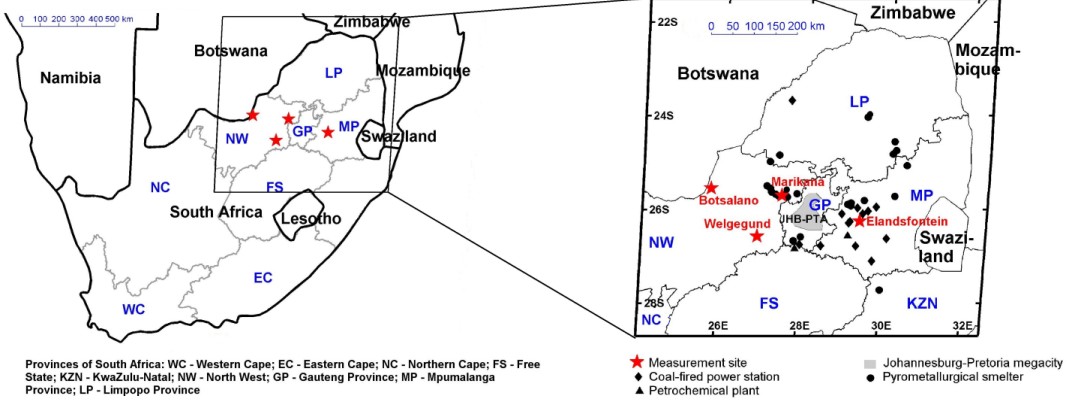

**Fig. 1.**    Location of the four measurement sites in South Africa.

### 2.1.1   Botsalano

Botsalano measurement site was situated in a game reserve in the North West province of South Africa, which is considered to be representative of regional background air. The surrounding vegetation is typical of a savannah biome, consisting of grasslands with scattered shrubs and trees (Laakso et al., 2008). The area is quite sparsely populated and has no local anthropogenic pollution sources (Laakso et al., 2008; Vakkari et al., 2013). The western Bushveld Igneous Complex where numerous platinum, base metal, vanadium and chromium mining/smelting industries are situated, is the largest regional anthropogenic pollution source with the Rustenburg area located approximately 150 km to the east. Botsalano is also occasionally impacted by plumes passing over the industrialised Mpumalanga Highveld and the Johannesburg-Pretoria megacity (Laakso et al., 2008; Vakkari et al., 2011). In addition, the site is influenced by seasonal regional savannah wildfires during the dry period (Laakso et al., 2008; Vakkari et al., 2011; Mafusire et al., 2016). Measurements were conducted from 20 July 2006 until 5 February 2008 (Laakso et al., 2008; Vakkari et al., 2011; Vakkari et al., 2013).

### 2.1.2   Marikana

The Marikana measurement site was located within the western Bushveld Igneous Complex, which is a densely populated and highly industrialised region, where mining and smelting are the predominant industrial activities. Marikana is a small mining town located about 30 km east



of Rustenburg and about 100 km northwest of Johannesburg. The measurement site was located in the midst of a residential area comprising low-cost housing settlements and municipal buildings (Hirsikko et al., 2012; Venter et al., 2012). Anthropogenic emissions from household combustion, traffic and industry in the wider region have a strong influence on the measurement site (Venter et al., 2012). Data was collected for a period from 8 February 2008 to 16 May 2010 and has been previously used in other studies (Venter et al., 2012; Vakkari et al., 2013; Petäjä et al., 2013; Hirsikko et al., 2012; Hirsikko et al., 2013).

### 2.1.3 Welgegund

This measurement site is approximately 100 km west of Johannesburg and is located on a commercial arable and pastoral farm. The station is surrounded by grassland savannah (Jaars et al., 2016). The station can be considered a regionally representative background site with few local anthropogenic sources. Air masses arriving at Welgegund from the west reflect a relatively clean regional background. However the site is, similar to the Botsalano station, at times impacted by polluted air masses that are advected over major anthropogenic source regions in the interior of South Africa, which include the western Bushveld Igneous Complex, Johannesburg-Pretoria megacity, the Mpumalanga Highveld and the Vaal Triangle (Tiitta et al., 2014; Jaars et al., 2016; Venter et al., 2017). In addition, Welgegund is also affected by regional savannah and grassland fires that are common in the dry season (Vakkari et al., 2014). The atmospheric measurement station has been operating at Welgegund since 20 May 2010 with data measured up until 31 December 2015 utilised in this study.

### 2.1.4 Elandsfontein

Elandsfontein is an ambient air quality monitoring station operated by Eskom, the national electricity supply company, primarily for legislative compliance purposes. This station was upgraded and co-managed by researchers during the EUCAARI project (Laakso et al., 2012). The Elandsfontein station is located within the industrialized Mpumalanga Highveld at the top of a hill approximately 200 km east of Johannesburg and 45 km south-southeast of eMalahleni (previously known as Witbank) which is a coal mining area (Laakso et al., 2012). The site is influenced by several emission sources such as coal mines, coal-fired power-generating stations, a large petrochemical plant and traffic emissions. Metallurgical smelters to the north also frequently impact the site (Laakso et al., 2012). The Elandsfontein data set covers the



period 11 February 2009 until 31 December 2010 during the EUCAARI campaign (Laakso et al., 2012).

## 2.2    Measurements

A comprehensive dataset of continuous measurements of surface aerosols, trace gases and meteorological parameters has been acquired through these four measurement sites (Laakso et al., 2008; Vakkari et al., 2011; Venter et al., 2012; Laakso et al., 2012; Vakkari et al., 2013; Petäjä et al., 2013). In particular, ozone ($O_3$), nitric oxide (NO), nitrogen dioxide ($NO_2$) and carbon monoxide (CO), as well as meteorological parameters, such as temperature ($^o$C) and relative humidity (%) measurements were used in this study. Note that Botsalano, Marikana and Welgegund measurements were obtained with the same mobile station (first located at Botsalano, then relocated to Marikana and thereafter permanently positioned at Welgegund), whilst Elandsfontein measurements were conducted with a routine monitoring station. $O_3$ concentrations at Welgegund, Botsalano and Marikana research stations were measured using the Environment SA 41M $O_3$ analyser, while a Monitor Europe ML9810B $O_3$ analyser was utilised at Elandsfontein. CO concentrations were determined at Welgegund, Botsalano and Marikana with a Horiba APMA-360 analyser, while CO was not measured at Elandsfontein. $NO_x$ ($NO+NO_2$) concentrations were determined with a Teledyne 200AU NO/$NO_x$ analyser at Welgegund, Botsalano and Marikana, whereas a Thermo Electron 42i NOx analyser was used at Elandsfontein. Temperature and relative humidity were measured with a Rotronic MP 101A instrument at all the sites.

Data quality at these four measurement sites were ensured through regular visits to the sites during which instrument maintenance and calibrations were performed. The data collected from these four stations were subjected to detailed cleaning (e.g. excluding measurements recorded during power interruptions, electronic malfunctions, calibrations and maintenance) and verification of data quality procedures (e.g. corrections were made to data according to in-situ calibrations and flow-checks). Therefore the datasets collected at all four measurement sites are considered to represent high quality, high resolution measurements as indicated by other papers (Laakso et al., 2008; Petäjä et al., 2013; Venter et al., 2012; 2011; Laakso et al., 2012; Vakkari et al., 2013). Detailed descriptions of the data post-processing procedures were presented by Laakso et al. (2008) and Venter et al. (2012). The data was available as 15-minute averages and all plots using local time (LT) refer to local South African time, which is UTC+2.



## 2.3 Air mass history

Individual hourly four-day back trajectories for air masses arriving at an arrival height of 100 m above ground-level were calculated for the entire measurement period at each monitoring site, using HYSPLIT 4.8 (Hybrid Single-Particle Lagrangian Integrated Trajectory model) (Stein et al., 2015; Draxler and Hess, 1998). The model was run with the GDAS meteorological archive produced by the US National Weather Service's National Centre for Environmental Prediction (NCEP) and archived by ARL (Air Resources Laboratory, 2017). Overlay back trajectory maps were generated by superimposing individual back trajectories onto southern African map divided into 0.5 X 0.5°grid cells. In addition, similar to the simple approach applied in other papers where concentrations of species were related to individual back trajectories (Vakkari et al., 2011; Vakkari et al., 2013; Tiitta et al., 2014), each grid cell was assigned the mean measured $O_3$ concentration associated with trajectories passing over that cell. A minimum of ten trajectories per cell was required for the statistical reliability.

## 2.4 Modelling instantaneous production rate of $O_3$

A mathematical model developed by Murphy et al. (2006) and used in Geddes et al. (2009) was applied to calculate the •OH radical concentration at a particular measurement time (Fig. A1). The production rate of $HO_x$ ($P(HO_x)$) was required to calculate the •OH radical concentration, which was estimated to be 0.89 ppb/h (calculated for an $O_3$ average of 41 ppbv and RH of 42 % at 11:00 LT each day). The factors and reactions that affect [•OH] include:

- linear dependency between •OH and $NO_x$ due to the reaction $NO + HO_2 \rightarrow \,•OH + NO_2$, until •OH begins to react with elevated $NO_2$ concentrations to form $HNO_3$ (OH + $NO_2$ + M→ $HNO_3$ + M);
- $P(HO_x)$ is affected by solar irradiance, temperature, $O_3$ concentrations, humidity; and
- partitioning of $HO_x$ between $RO_2$, $HO_2$, OH.

[•OH] was calculated at 11:00 LT each day as follows:

$$A = k_{5eff}(\frac{VOC\ reactivity}{k_{2eff}[NO]})^2$$
$$B = k_4[NO_2] + \alpha * VOC\ reactivity$$
$$C = P(HO_x)$$



$$[OH] = \frac{-B + \sqrt{B^2 + 24C*A}}{12*A}$$

The instantaneous production rate of $O_3$, $P(O_3)$ could then be calculated as a function of $NO_2$ levels and VOC reactivity. VOC reactivity was calculated from the product of the VOC

concentration and its rate constant for the reaction with a $^\bullet OH$ radical (Seinfeld and Pandis, 2006), which were then summed to obtain the total VOC reactivity for each measurement, i.e. VOC reactivity = $\sum k_{1i}[VOC]_i$ (Fig. A2). A set of reactions used to derive the equations that describe the dependence of the $^\bullet OH$, peroxy radicals ($HO_2^\bullet + RO_2^\bullet$) and $P(O_3)$ on $NO_x$ is given by Murphy et al. (2006), which present the following equation to calculate $P(O_3)$:

$$P(O_3) = k_{2eff}[HO_2 + RO_2][NO] = 2k_1 * VOC\ Reactivity * [OH]$$

where $k_1$ is the rate constant of VOC oxidation by $^\bullet OH$; $k_2eff$ is the effective rate constant of NO oxidation by peroxy radicals (chain propagation and -termination reactions in the production of

$O_3$). The values of the rate constants and other parameters e.g. concentrations of peroxy radicals and the hydroxyl radical used as input parameters to solve the equation above, can be found in Murphy et al. (2006) and Geddes et al. (2009).

### 3.    Results and discussion

### 3.1    Contextualisation of $O_3$ levels

### 3.1.1    Spatial distribution of $O_3$ in continental South Africa

In order to contextualise the $O_3$ concentrations measured at the four sites in this study and to obtain a representative spatial coverage of continental South Africa, $O_3$ data from an additional 54 ambient monitoring sites was selected. This included $O_3$ measurements from 18 routine monitoring stations measurements (SAAQIS) for the period Jan 2012 – Dec 2014 (downloaded from the JOIN web interface https://join.fz-juelich.de (Schultz et al., 2017)) and 36 passive

sampling sites located in the north-eastern interior of South Africa where monthly $O_3$ concentrations were determined for two years from 2006 to 2007 (Josipovic, 2009). Spatial analysis were conducted with a geographical information system mapping tool (ArcGIS





software), which used ordinary kriging to interpolate the $O_3$ concentrations measured at the 58 sites in order to build the spatial distribution. The interpolation method involved making an 80/20% split of the data (80% for model development, 20% for evaluation) where 20% was used to calculate the root mean squared error (RSME = 0.2804331). Optimal model parameters were

selected using an iterative process and evaluated on the basis of the best performance statistics obtained (reported in the ArcGIS kriging output), with particular emphasis on minimising the RSME. The extent of area was 23.00154974 (top), -29.03070026 (bottom), 25.74238974 (left) and 32.85246366 (right). Figure 2 depicts the spatial pattern of mean surface $O_3$ concentrations over continental South Africa during springtime (S-O-N), when $O_3$ is usually at a maximum as

indicated above.

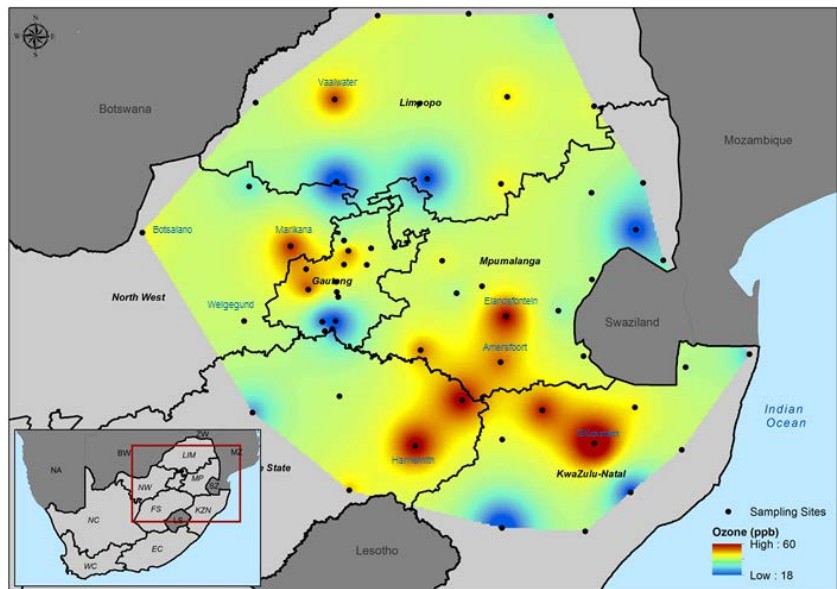

**Fig. 2.**      Spatial distribution map of mean surface $O_3$ levels during springtime over the north-eastern interior of southern Africa ranging between 23.00 ° S and 29.03 ° S, and 25.74 ° E and 32.85 ° E. The data for all sites were averaged for years when the ENSO cycle was not present (by examining SST anomalies in the Niño 3.4 region). Black dots indicate the sampling sites.

The mean $O_3$ concentration over continental South Africa ranged from 20 ppb to 60 ppb during spring. From Fig. 2 it can be seen that $O_3$ concentrations at the industrial sites Marikana and



Elandsfontein were higher than O$_3$ levels at Botsalano and Marikana. As mentioned previously, Elandsfontein is located within the industrialized Mpumalanga Highveld with numerous large point sources of O$_3$ precursor species. It is also evident from Fig. 2 that rural measurement sites downwind from Elandsfontein, such as Amersfoort, Harrismith and Glückstadt had significantly

higher O$_3$ concentrations, which can be attributed to the formation of O$_3$ during transport of precursor species from source regions. Lourens et al. (2011) indicated that higher O$_3$ concentrations were associated with sites positioned in more rural areas in the Mpumalanga Highveld. Venter et al. (2012) attributed high O$_3$ concentrations at Marikana, which exceeded South African standard limits on a number of occasions, to the influence of local household

combustion for cooking and space heating, as well as to regional air masses with high O$_3$ precursor concentrations. Higher O$_3$ concentrations were also measured in the north-western parts of Gauteng at sites situated within close proximity of the Johannesburg-Pretoria megacity, while the rural Vaalwater site in the north also has significantly higher O$_3$ levels. From Fig. 2 it is evident that O$_3$ can be considered a regional problem with O$_3$ concentrations being higher than

40 ppb across continental South Africa during spring. Figure 2 also clearly indicates that the four research sites where surface O$_3$ was assessed in this study are representative of continental South Africa.

### 3.1.2  Comparison with international sites

In an effort to contextualise O$_3$ concentrations measured at South African sites, the monthly O$_3$ concentrations measured in South Africa were compared to monthly O$_3$ levels measured at monitoring sites in other parts of the world (downloaded from the JOIN web interface https://join.fz-juelich.de (Schultz et al., 2017)) as indicated in Fig. 3. The measurement time

period considered was from May 2010 to December 2014. Of the four sites assessed in this paper, only Welgegund was used in the comparison since it had the most extensive data record, while three other South African routine monitoring stations influenced by local and regional pollution sources were included, i.e. Hendrina (rural), Sebokeng (industrial, low-income housing) and Witbank (industrial) (SAAQIS). It is evident from Fig. 3 that the rural sites with few local

sources (Welgegund, Hendrina) experience higher O$_3$ levels than Witbank and Sebokeng, which are situated in highly industrialised and densely populated urban areas. The seasonal O$_3$ cycles at Hendrina, Witbank and Sebokeng are similar to that observed at Welgegund. The seasonal O$_3$ cycles observed at other sites in the Southern Hemisphere are comparable to the seasonal cycles at the South African sites with slight variations in the time of year when O$_3$ peaks as



indicated in Fig. 3. Cape Grim, Australia; GoAmazon T3 Manancapuru, Brazil; Ushuaia, Argentina; and El Tololo, Chile are regional background GAW (Global Atmosphere Watch) stations with $O_3$ levels lower than the South African sites. However, the $O_3$ concentrations at El Tololo, Chile are comparable to Welgegund. Oakdale, Australia and Mutdapliiy, Australia are

semi-rural and rural locations, which are influenced by urban and industrial pollution sources, which also had lower $O_3$ concentrations compared to the South African sites.

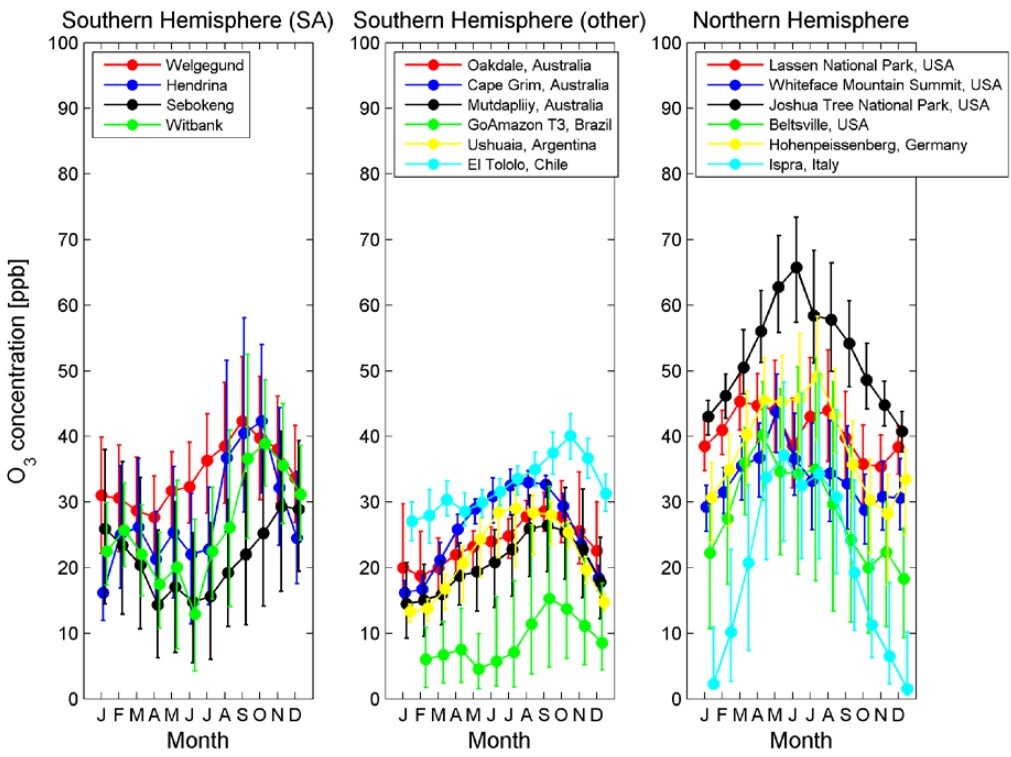

**Fig. 3.**      Seasonal cycle of $O_3$ at rural sites in other parts of the world. The black dot indicate monthly median (50th percentile) and the upper and lower limits the 25th and 75th percentile, respectively for monthly $O_3$ concentrations. The data is averaged from May 2010 to December 2014, except in a few instances where 2014 data was not available.

The Northern Hemispheric $O_3$ peak over mid-latitude regions is similar to seasonal patterns in the Southern Hemisphere where a springtime $O_3$ maximum is observed (e.g. Whiteface



Mountain Summit) although there are other sites in the Northern Hemisphere where a summer maximum is more evident (Vingarzan, 2004), such as Joshua Tree and Beltsville. The spring maximum in the Northern Hemisphere is associated with stratospheric intrusions (Zhang et al., 2014; Parrish et al., 2013), while the summer maximum is associated with photochemical $O_3$
production from anthropogenic emissions of $O_3$ precursors being at its highest (Logan, 1985; Chevalier et al., 2007). Maximum concentrations in the United States and Europe are similar to values at the South African sites in spring. The exceptions are Lassen National Park and Joshua Tree National Park in the United States, which have higher values than the Southern Hemispheric sites. Note that Lassen National Park is at a slightly higher altitude than the South
African sites and, generally, $O_3$ concentrations increase with increasing elevation (Jaffe and Ray, 2007; Burley and Bytnerowicz, 2011) as is evident. Joshua Tree National Park shows the highest $O_3$ levels from all sites, reaching monthly means between 60 and 70 ppb during summer, most likely due to its high elevation and deep boundary layer (~4 km asl) during spring and summer allowing free tropospheric $O_3$ to be more effectively mixed down to the surface
(Cooper et al., 2014). Similarly in Europe, there is either a spring maximum as observed at Hohenpeissenberg or a summer maximum as seen at Ispra. The latter has similar $O_3$ levels during spring and summer as the South African sites, but decreases significantly during the rest of the year. The discernible difference between the hemispheres is that the spring maximum in the Southern Hemisphere refers to maximum $O_3$ concentrations in late winter and early spring,
whilst in the Northern Hemisphere it refers to a late spring and early summer $O_3$ maximum (Cooper et al., 2014).

### 3.2    Seasonal and diurnal variation of $O_3$

In Fig. 4 the monthly and diurnal variation for $O_3$ concentrations measured at the four research sites in this study are presented. Although there is some variability between the sites, monthly $O_3$ concentrations show a well-defined seasonal variation at all four sites, with maximum concentrations occurring in late winter and spring (August-November). This observed late winter and spring $O_3$ peak are expected for the South African interior as previously reported (Zunckel
et al., 2004; Diab et al., 2004; Combrink et al., 1995). These $O_3$ peaks in continental South Africa, generally points to two major contributors of $O_3$ precursors, i.e. open biomass burning (wild fires) (Fishman and Larsen, 1987; Vakkari et al., 2014) and increased low-level anthropogenic emissions (Oltmans et al., 2013; Lourens et al., 2011). In addition, not only are some $O_3$ precursor sources seasonal, but during the dry winter months synoptic scale





recirculation is more predominant and inversion layers are more pronounced, while precipitation is minimal (e.g. Tyson and Preston-Whyte, 2000). $O_3$ concentrations increase in the dry period due to the build-up of precursor species and reach a maximum in August/September when photochemical activity starts to increase. The influences of sources of $O_3$ precursors will be explored in Section 3.3.

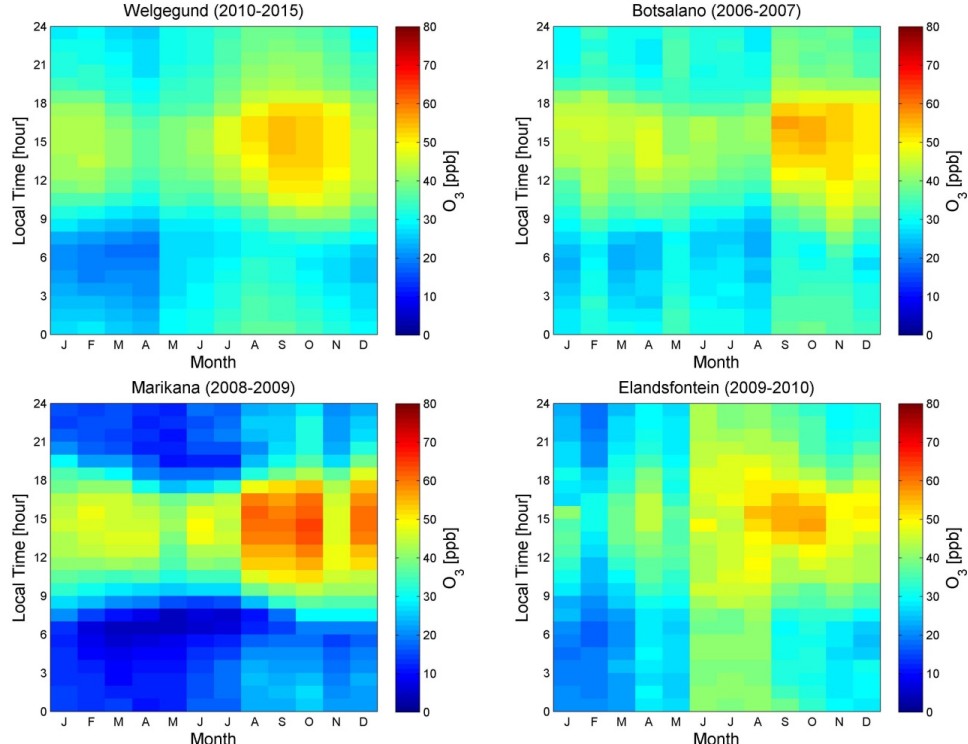

**Fig. 4.** Seasonal and diurnal variation of median $O_3$ concentrations at Welgegund, Botsalano, Marikana and Elandsfontein. The $O_3$ measurement periods varied among sites, which combined spanned a period from July 2006 to December 2015.

The diurnal concentration profiles of $O_3$ at the four locations follow the photochemical cycle, i.e. decreasing during the nighttime and increasing during daytime. $O_3$ peaked from midday to afternoon, with a maximum at approximately 15:00 (LT, UTC+2), in response to maximum photochemical production (Seinfeld and Pandis, 1998; Crutzen et al., 1999). In the absence of solar radiation during nighttime, photochemical production of $O_3$ ceases and titration of $O_3$ occurs due to reaction with NO resulting in higher $NO_2$ (Eq. (3)) (Dueñas et al., 2002). In



addition, O$_3$ is also removed through dry deposition at night. The lower O$_3$ levels are maintained throughout the night and early morning hours with a minimum just before sunrise at approximately 6:00 LT. After sunrise, the inversion layer gradually breaks up and the O$_3$ concentrations steadily start to increase due to the downward mixing of the residual layer and

increasing sunshine. O$_3$ formation starts later in the day (around 7:00 LT) in autumn and winter due to the shift in local time of sunrise. From Fig. 4 is also evident that nighttime titration of O$_3$ at Marikana is more pronounced as indicated by the largest difference between daytime and nighttime O$_3$ concentrations in comparison to the other sites, especially, compared to Elandsfontein where nighttime concentrations of O$_3$ remain relatively high in winter.

### 3.3    Sources contributing to surface O$_3$ in continental South Africa

As indicated above, the O$_3$ peaks in continental South Africa usually reflects increased concentrations of precursor species from anthropogenic sources during winter, as well as the

occurrence of regional open biomass burning in late winter and early spring. In addition, stratospheric O$_3$ intrusions during the spring (Lefohn et al., 2014) could also partially contribute to increased surface O$_3$ levels.

#### 3.3.1    Anthropogenic and open biomass burning emissions

Comparison of the O$_3$ seasonal cycles at background and polluted locations is useful for source attribution. In Fig. 5 the monthly average daytime (11:00-17:00 LT) O$_3$ concentrations at the four sites are compared. Daytime measurements were used, since the boundary layer height was high and well-mixed, while nighttime surface deposition did not occur (Cooper et al., 2012).

From Fig. 5 it is evident that daytime O$_3$ levels peaked at Elandsfontein, Marikana and Welgegund during late winter and spring (August to October), while O$_3$ levels at Botsalano peaked later in the year during spring (September to November). Therefore Elandsfontein, Marikana and Welgegund were influenced by increased levels of O$_3$ precursors from anthropogenic and open biomass burning emissions (i.e. NO$_x$ and CO indicated in Fig. A3 and

Fig. A4 respectively), while O$_3$ levels at Botsalano were predominantly influenced by regional open biomass burning (Fig. A4). As mentioned previously, during winter an increase in concentration atmospheric pollutants occurs in continental South Africa due to increased household combustion for space heating, as well as the prevailing meteorological conditions, i.e. more pronounced anticyclonic recirculation and inversion layers causing decreased vertical





mixing. High $O_3$ levels in spring can also be related to increased local photochemical production of $O_3$ in conjunction with increased concentrations of precursors (Atlas et al., 2003; Carvalho et al., 2010). Although Welgegund and Botsalano are both background sites, Botsalano is more removed from anthropogenic source regions than Welgegund (Section 2.1.3), which is therefore
5   not directly influenced by the increased concentrations of $O_3$ precursor species associated with anthropogenic emissions during winter. Daytime $O_3$ concentrations were the highest at Marikana throughout most of the year, which indicate the influence of local and regional sources of $O_3$ precursors at this site (Venter et al., 2012).

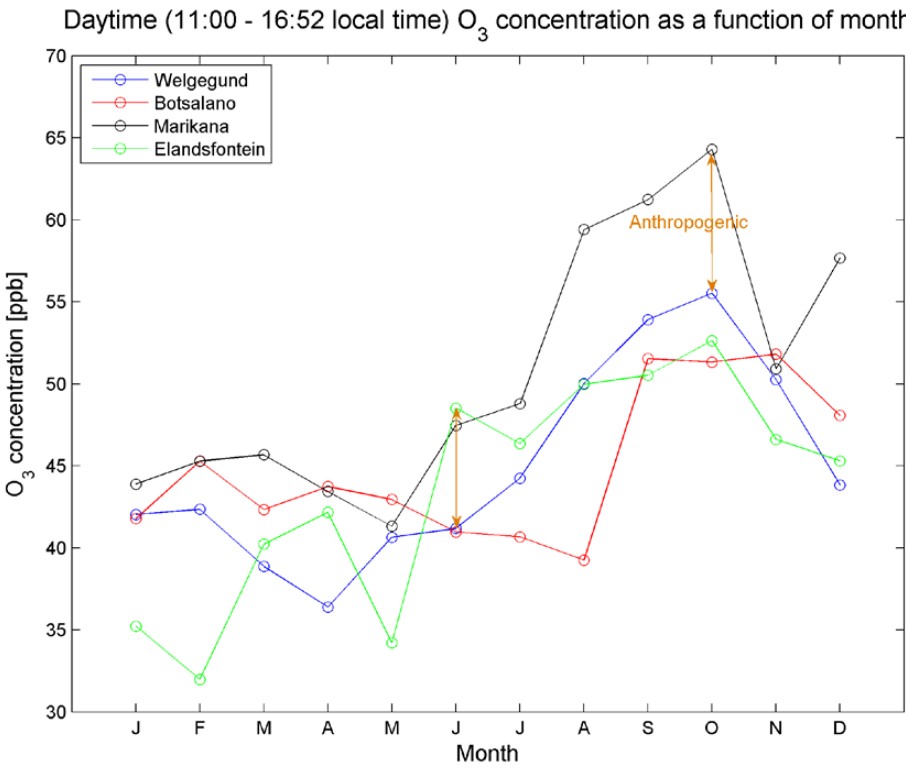

**Fig. 5.**      Monthly mean daytime (11:00 to 17:00 LT) $O_3$ levels at the four continental sites in South Africa. The $O_3$ measurement periods varied among sites, which combined spanned a period from July 2006 to December 2015.

15   $O_3$ concentrations at Elandsfontein were lower compared to the other three sites throughout the year, with the exception of the winter months (June to August). Although major point sources





impacting Elandsfontein are important sources of $O_3$ precursors (e.g. $NO_x$ emissions from coal-fired power stations), these point sources are characterized by high-stack emissions, which are emitted above the low-level nighttime inversion layers. During daytime downwards mixing of these emitted species occurs, which result in daytime peaks of $NO_x$ (as indicated in Fig. A5 and

by Collett et al., 2010) and subsequent $O_3$ titration. In contrast, Venter et al. (2012) indicated that at Marikana low-level emissions (below the nighttime inversion layer) associated with household combustion for space heating and cooking was a significant source of $O_3$ precursor species, i.e. $NO_x$ and CO. The diurnal pattern of $NO_x$ and CO (Fig. A3 and Fig. A4 respectively) at Marikana was characterised by bimodal peaks during the morning and evening, which

resulted in increased $O_3$ concentrations during daytime and nighttime titration of $O_3$, especially during winter. Therefore the observed differences in nighttime titration at Marikana and Elandsfontein can be attributed different sources of $O_3$ precursors, i.e. mainly low-level emissions (household combustion) at Marikana compared to predominant high-stack emissions at Elandsfontein. The higher $O_3$ concentrations at Elandsfontein during winter are most-likely

attributed to the regional increase in $O_3$ precursors.

The influence of anthropogenic emissions on $O_3$ concentrations can also be illustrated by comparison of the monthly $O_3$ levels measured at the rural background and an industrial site. Comparison between $O_3$ concentrations at Welgegund and Marikana indicated small differences

between the $O_3$ levels during the summer (January, February) and autumn (March to May) months. During June a ~10 ppb monthly $O_3$ concentration difference is observed, which increases to ~15 ppb during October. This baseline shift observed between Welgegund and Marikana can be attributed to local anthropogenic emissions (mainly household combustion) of $O_3$ precursors at Marikana. The baseline shift is smaller in winter due to increased $O_3$ titration

associated with increased $NO_x$ emissions.

The spring maximum $O_3$ concentrations can be attributed to increases in widespread regional biomass burning in this region during this period (Vakkari et al., 2014; Lourens et al., 2011). Biomass burning has strong seasonality in southern Africa, extending from June to September

(Galanter et al., 2000), and is an important source of $O_3$ and its precursors during the dry season. In an effort to elucidate the influence of regional biomass burning on $O_3$ concentrations in continental South Africa, source area maps of $O_3$ were compiled by relating $O_3$ concentrations measured with air mass history, which are presented in Fig. 6 (a). Source area maps were only generated for the background sites Welgegund and Botsalano, since local sources at the



industrial sites Elandsfontein and Marikana would obscure the influence of regional biomass burning. In addition, maps of spatial distribution of fires during 2007, 2010 and 2015 were compiled with the MODIS collection 5 burnt area product (Roy et al., 2008; Roy et al., 2005; Roy et al., 2002), which are presented in Fig. 7.

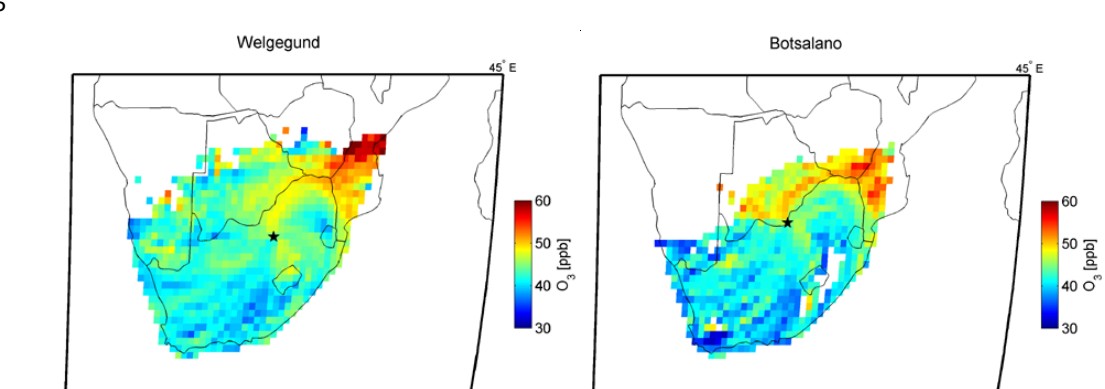

10 **(a)**

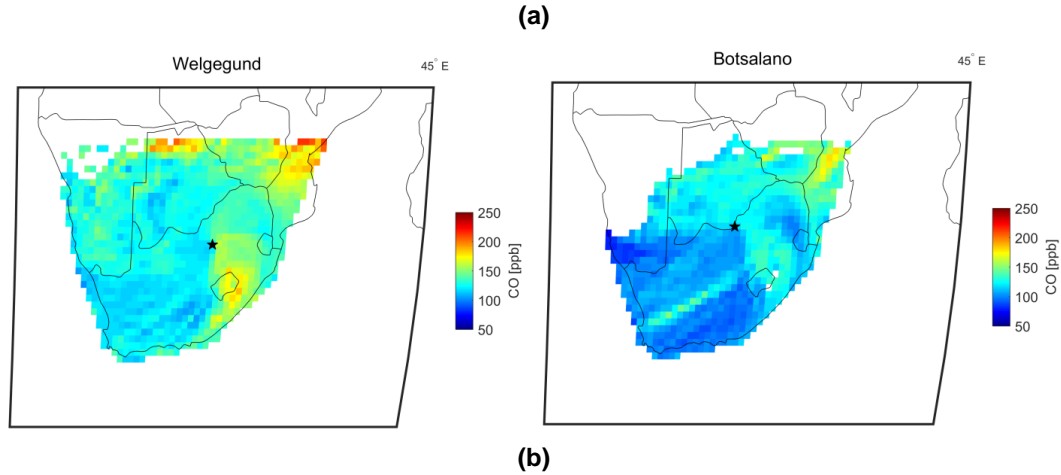

**(b)**

15 **Fig. 6.** Source area maps of (a) $O_3$ concentrations and (b) CO concentrations for the background sites Welgegund and Botsalano. The black star represents the measurement site and the colour of each pixel represents the mean concentration of the respective gas species. At least ten observations per pixel are required.




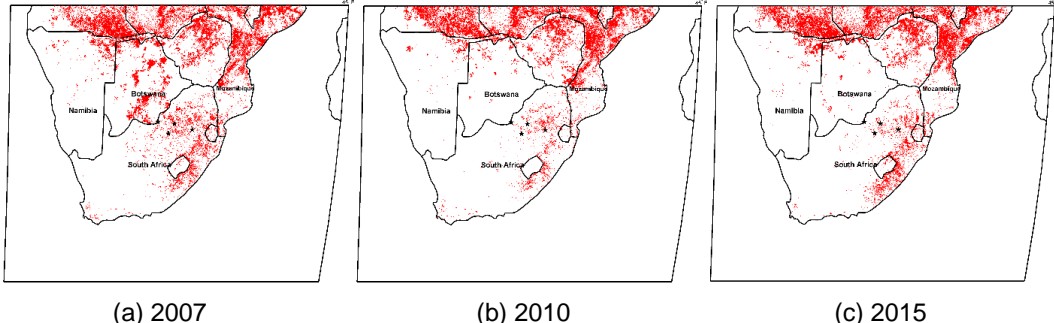

(a) 2007          (b) 2010          (c) 2015

**Fig. 7.**    Spatial distribution of fires in 2007, 2010 and 2015 from MODIS burnt area product. Black stars indicate (from left to right) Botsalano, Welgegund, Marikana and Elandsfontein.

The highest $O_3$ concentrations measured at Welgegund and Marikana were associated with air masses passing over a sector north to north-east of these sites, i.e. southern and central Mozambique, southern Zimbabwe and south-eastern Botswana. $O_3$ concentrations associated with air masses passing over central and southern Mozambique were particularly high. In addition to $O_3$ source maps, CO source maps were also compiled for Welgegund and Botsalano as indicated in Fig. 6 (b). It is evident that the CO source maps indicated a similar pattern than that observed for $O_3$ with the highest CO concentrations corresponding with the same regions where $O_3$ levels are the highest. From the fire maps in Fig. 7 it can be observed that a large number of fires occur in the sector associated with higher $O_3$ and CO concentrations, with the fire map indicating, especially, high fire frequency occurring in central Mozambique. During 2007 more fires occurred in Botswana compared to the other two years, which is also reflected in the higher $O_3$ levels measured at Botsalano during that year for air masses passing over this region. Open biomass burning is known to emit more CO than $NO_x$, while CO also has a relatively long atmospheric lifetime (1 to 2 months, Kanakidou and Crutzen, 1999) compared to $NO_x$ (6 to 24 hours, Beirle et al., 2003) and VOCs (few hours to a few weeks, Kanakidou and Crutzen, 1999) emitted from open biomass burning. Enhanced CO concentrations have been used previously to characterise the dispersion of biomass burning emissions over southern Africa (Mafusire et al., 2016). Therefore the regional transport of CO (and $NO_x$ and VOCs to a lesser extent) associated with biomass burning occurring from June to September in southern Africa can be considered an important source of surface $O_3$ in continental South Africa (Fig. A4).





### 3.3.2 Stratospheric $O_3$

Elevated levels of tropospheric $O_3$ may also be caused by stratospheric intrusion of $O_3$-rich air (Zhang et al., 2014; Parrish et al., 2013; Lin et al., 2012), especially on certain days during late

winter and spring when $O_3$ is the highest on the South African Highveld (Thompson et al., 2014). However, the importance of the stratospheric source over continental South Africa has not yet been specifically addressed. Assessment of meteorological fields and air quality data at high-elevation sites is required to determine the downward transport of stratospheric $O_3$. Alternatively, stratospheric $O_3$ intrusions can be estimated through concurrent in situ

measurements of ground level $O_3$, CO and humidity, since stratospheric intrusions of $O_3$ into the troposphere are characterised by elevated levels of $O_3$, high potential vorticity, low levels of CO and low water vapour (Stauffer et al., 2017; Thompson et al., 2015; Thompson et al., 2014). Thompson et al. (2015) defined low CO as 80 to 110 ppbv, whilst low relative humidity (RH) is considered <15 %. In Fig. 8 the 95th percentile $O_3$ levels (indicative of "high $O_3$") corresponding

to low daily average CO concentrations (< 100 ppb) are presented together with the daily average RH. Only daytime data from 07:00-18:00 (LT) were considered in order to exclude the influence of nighttime titration. From Fig. 8 it is evident that very few days complied with the criteria indicative of stratospheric $O_3$ intrusion, i.e. high $O_3$, low CO and low RH, which indicates very small influence of stratospheric intrusion on surface $O_3$ levels. However, it must be noted

that the attempt in this study to related surface $O_3$ to stratospheric intrusions is a simplified qualitative assessment and more quantitative detection methods should be applied to understand the influence of stratospheric intrusions on surface $O_3$ for this region.



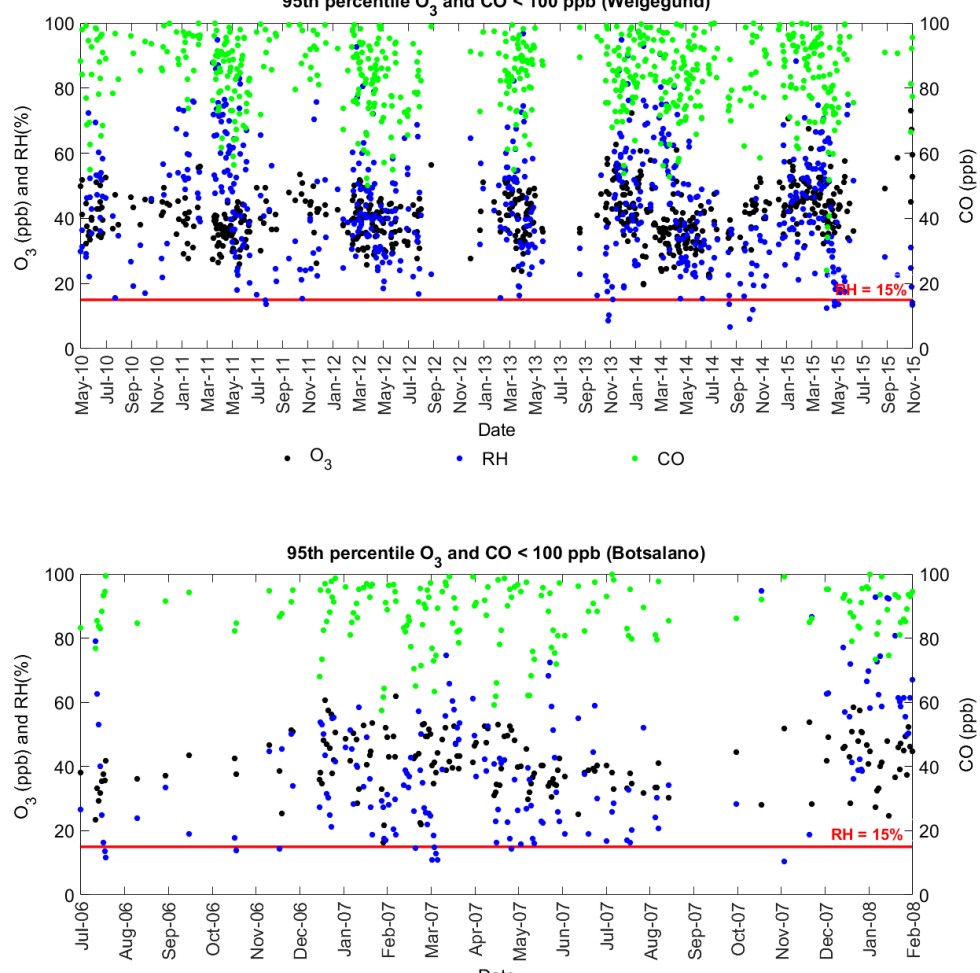

**Fig. 8.** Simultaneous measurements of $O_3$ (daily 95[th] percentile), CO (daily average ppb) and RH (daily average) from 07:00 to 18:00 LT at Welgegund, Botsalano and Marikana.



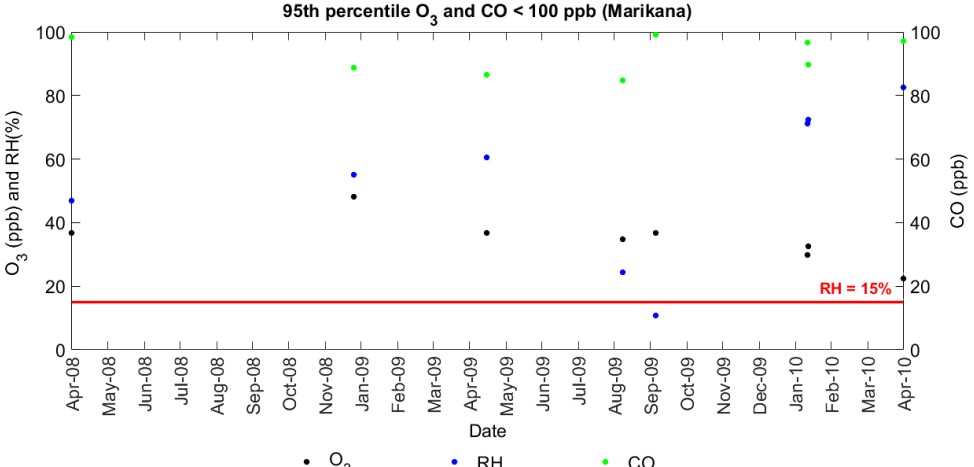

**Fig. 8.**     Continued.

### 3.4    Insights into the O₃ production regime

In the absence of VOC data, the relationship between $O_3$, $NO_x$ and CO was used as an indicator to infer the $O_3$ production regime at Welgegund, Botsalano and Marikana (no CO measurements were conducted at Elandsfontein as indicated above). A two-year VOC dataset compiled during two sampling campaigns was available for Welgegund (Jaars et al., 2016; Jaars et al., 2014), which was used to calculate the instantaneous production rate of $O_3$ as a function of $NO_2$ levels and VOC reactivity (Geddes et al., 2009; Murphy et al., 2006).

#### 3.4.1    The relationship between NOₓ, CO and O₃

In Fig. 9 the correlations between $O_3$, $NO_x$ and CO concentrations at Welgegund, Botsalano and Marikana are presented, which clearly indicates higher $O_3$ concentrations associated with increased CO levels, while $O_3$ levels remain relatively constant (or decrease) with increasing $NO_x$. The highest $O_3$ concentrations occur for $NO_x$ levels below 10 ppb, since the equilibrium between photochemical production of $O_3$ and chemical removal of $O_3$ shifts towards the former i.e. greater $O_3$ formation. In general there seems to exist a marginal negative correlation between $O_3$ and $NO_x$ (Fig. A5) at all four sites, which is a reflection of the photochemical production of $O_3$ from $NO_2$ (Eq. (1) and Eq. (2)) and the destruction of $O_3$ through $NO_x$ titration





(Eq. (3)). These correlations between $NO_x$, CO and $O_3$ indicate that $O_3$ production in continental South Africa is limited by CO concentrations, i.e. VOC(CO)-limited.

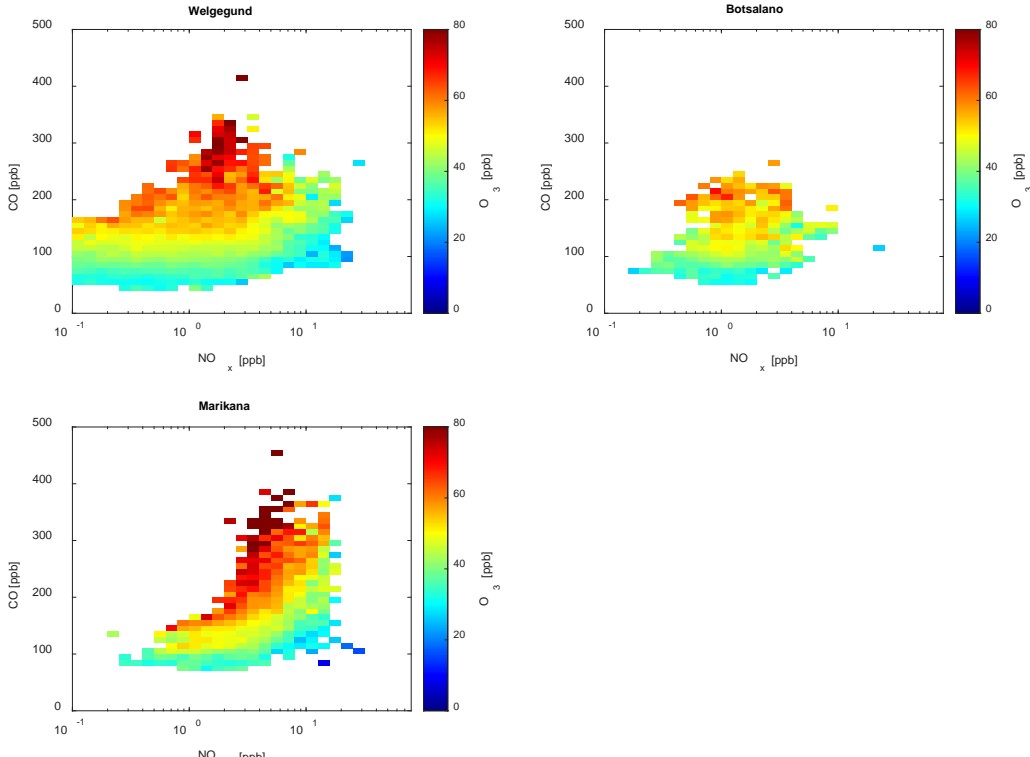

**Fig. 9.** Mean $O_3$ concentration averaged for $NO_x$ and CO bins. Measurements were only taken during period 11:00-17:00 LT when photochemical production of $O_3$ is at a maximum.

This finding shows a strong dependence of $O_3$ on CO and suggests that high $O_3$ can be mainly

10  attributed to oxidation of CO in the air masses, i.e. as long as there is sufficient amount of $NO_x$ present in a region, CO serves to produce $O_3$. Although $NO_x$ and VOCs are usually considered as the main precursors in ground-level $O_3$ formation, CO acts together with $NO_x$ and VOCs in the presence of sunlight to drive photochemical $O_3$ formation (Eq. (5)). According to Fig. 9, reducing CO emission should result in a reduction in surface $O_3$ and it is assumed that this

15  response is analogous to that of VOCs. It is, however, not that simple since the ambient $NO_x$ and VOCs concentrations are directly related to the instantaneous rate of production of $O_3$ and not necessarily to the ambient $O_3$ concentration at a location, which is the result of chemistry, deposition and transport that has occurred over several hours or a few days (Sillman, 1999).





Notwithstanding the various factors contributing to increased surface $O_3$ levels, the correlation between ambient CO and $O_3$ is, especially, relevant given the low reactivity of CO with respect to $^\bullet$OH radicals compared to most VOCs, which implies that the oxidation of CO probably takes place over a timescale of several days. It seems that the role of CO is of major importance in

tropospheric chemistry in this region where sufficient $NO_x$ is present across continental South Africa and biogenic VOCs are relatively less abundant (Jaars et al., 2016), to fuel the $O_3$ formation process.

### 3.4.2   Seasonal change in $O_3$-precursors relationship

Seasonal changes in the relationship between $O_3$ and precursor species can be indicative of different sources of precursor species during different times of the year. In Fig. 10 the correlations between $O_3$ levels with $NO_x$ and CO are presented for the different seasons, which indicate seasonal changes in the dependence of elevated $O_3$ concentrations on these

precursors. The very high CO concentrations relative to $NO_x$ i.e. high CO to $NO_x$ ratios are associated with the highest $O_3$ concentrations, which is most pronounced (highest $CO/NO_x$ ratios) during winter and spring. This indicates that the winter and spring $O_3$ maximum is primarily driven by increased $HO_2^\bullet$ production from CO (Eq. (5)). The seasonal maximum in $O_3$ concentration coincides with the maximum CO concentration at the background sites, whilst the

$O_3$ peak occurs just after June/July when CO peaked at the polluted site Marikana (Fig. A4). This observed seasonality in $O_3$ production signifies the importance of CO to $O_3$ formation in continental South Africa, as well as that CO emissions from open biomass burning during winter and spring can be considered to be a major source of $O_3$ in this region, while household combustion for space heating and cooking is also an important source of CO during winter as

previously discussed. The strong diurnal CO concentration patterns observed during winter at Marikana (Fig. A4) substantiate the influence of household combustion on CO levels as indicated by Venter et al. (2012).







**Fig. 10.** Seasonal plots of the relationship between $O_3$, $NO_x$ and CO at Welgegund, Botsalano and Marikana.

### 3.4.3 $O_3$ production rate

A VOC dataset was available from two sampling campaigns conducted at Welgegund from 2014 to 2016 (Jaars et al., 2016), which was used to calculate VOC reactivity and $P(O_3)$ as described in Section 2.4 at this site. The $P(O_3)$ at Welgegund as a function of VOC reactivity


and $NO_2$ concentrations is presented in Fig. 11. The dashed black line in Fig. 11, often called the ridge line, separates the $NO_x$- and VOC-limited regimes. To the left of the ridge line is the $NO_x$-limited regime, i.e. $O_3$ concentrations decrease with decreasing $NO_x$ and are insensitive to VOCs, while to the right of the ridge is the VOC-limited regime, i.e. $O_3$ concentrations decrease

5   with decreasing VOCs and increase with decreasing $NO_x$. According to the $O_3$ production plot presented, at least 40% of the data is found in the VOC-limited regime area. However, the $O_3$ production plot for Welgegund also indicates a $NO_x$-limited region at very low $NO_x$ concentrations (<1 ppb). Therefore Welgegund transitions between $NO_x$- and VOC-limited regimes. This can be attributed to Welgegund being impacted by the major source regions in

10  the north-eastern interior of South Africa, as well as a relatively clean background region (Tiitta et al., 2014). Therefore, for clean background air $O_3$ production is most-likely $NO_x$-limited, while large parts of the regional background of continental South Africa can be considered VOC-limited.

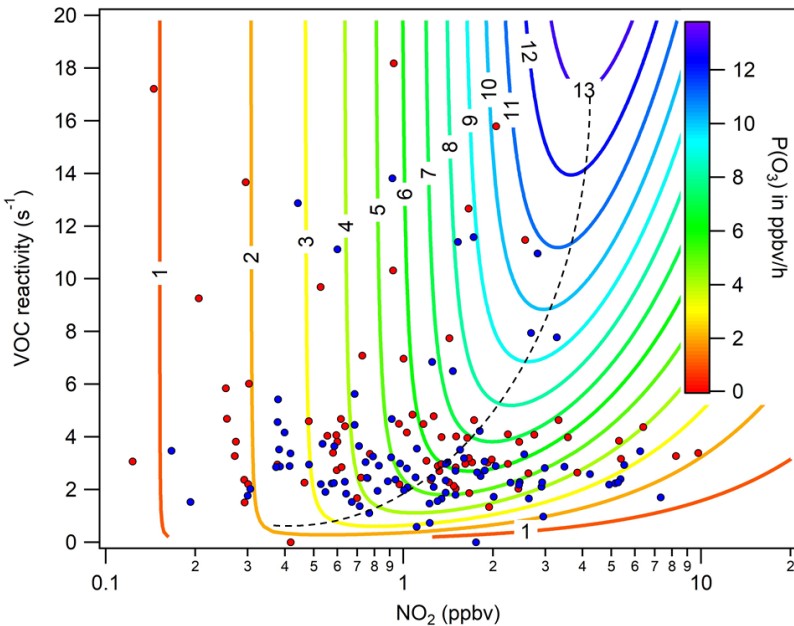

**Fig. 11.**    Contour plot of instantaneous $O_3$ production ($P(O_3)$) at Welgegund using daytime (11:00 LT) surface measurements of VOCs and $NO_2$ based on the model developed by Murphy et al. (2006). The blue dots represent the first campaign (2011-2012), and the red dots indicate the second campaign (2014-2015).



## 3.5 Implications for air quality management

### 3.5.1 Ozone exceedances

The South African National Ambient Air Quality Standard (NAAQS) for $O_3$ is an 8-hour moving average limit of 61 ppbv with 11 exceedances allowed annually (Government Gazette Republic of South Africa, 2009). Figure 12 shows the average number of days per month when this $O_3$ standard limit was exceeded at the four measurement sites. It is evident that the daily 8-h-$O_3$-maximum concentrations regularly exceeded the NAAQS threshold for $O_3$ and the number of

exceedances annually allowed at all the sites, including the most remote of the four sites, Botsalano. At the polluted locations of Marikana and Elandsfontein, the $O_3$ exceedances peak early on in the dry season (June onwards), whilst at the background locations of Welgegund and Botsalano, the highest number of exceedances occur later in the dry season (August to Nov). These relatively high number of $O_3$ exceedances at all the sites (background and

industrial) highlights the regional $O_3$ problem in South Africa, with background sites being impacted by the regional transport of $O_3$ precursors from anthropogenic and biomass burning source regions.



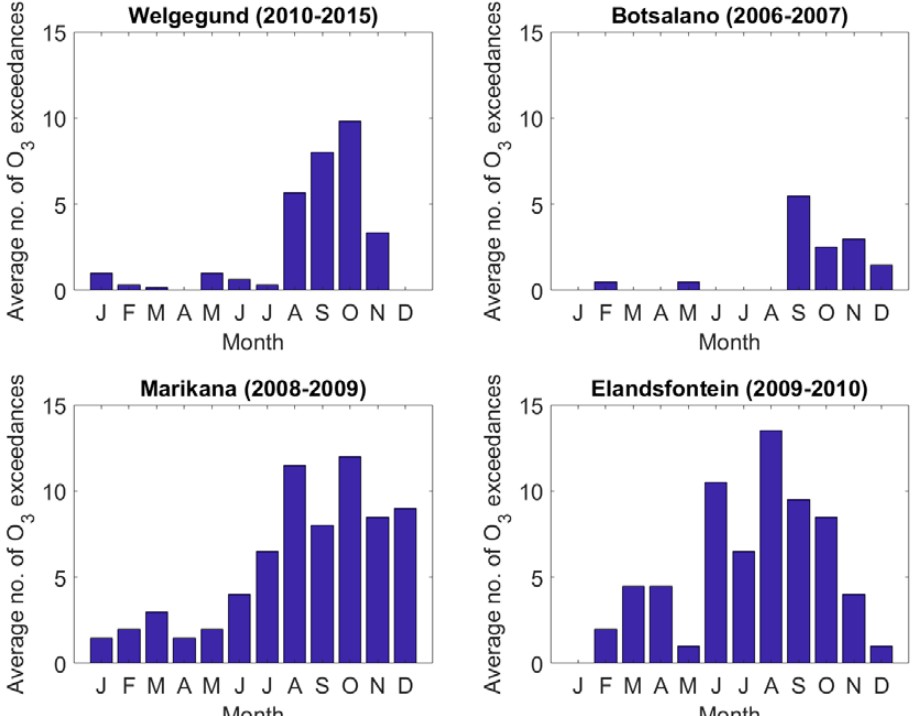

**Fig. 12.** Monthly number of exceedances of the daily 8-h-$O_3$-max (i.e. highest value of all available 8-hour moving averages in that day) above 61 ppbv at Welgegund, Botsalano, Marikana and Elandsfontein.

### 3.5.2 $O_3$ control strategies

The inverse relationship between $O_3$ and $NO_x$ at all four study sites (Fig. A5) is evidence that photochemical $O_3$ production is inhibited by high $NO_x$. In addition, the colour map of $O_3$ concentration as a function of $NO_x$ and CO concentrations at Welgegund, Botsalano and Marikana (Fig. 9), shows that high $O_3$ depends strongly on high CO, while the contour plot of $P(O_3)$ as a function of VOC reactivity and $NO_x$ at Welgegund (Fig. 11) also indicated a relatively strong dependence on VOCs. Therefore, the regions where Welgegund, Botsalano and Marikana are located can be considered VOC(CO)-limited. It can even be deducted that these sites are CO-limited, although this term "CO-limited" is not commonly used when referring to $O_3$ production regimes (Seinfeld and Pandis, 2006). In addition, since Elandsfontein is located in a highly industrialised region with high $NO_x$ emission, this area could also be considered VOC-limited. Rural remote regions are generally considered to be $NO_x$-limited due to the impact of



BVOCs and the availability of $NO_x$ that limits photochemical $O_3$ formation (Sillman, 1999). However, Jaars et al. (2016) indicated that BVOC concentrations at a savannah-grassland were at least an order of magnitude lower compared to other regions in the world, which together with high anthropogenic emissions of $NO_x$ in the interior of South Africa result in VOC-limited

conditions at background sites in continental South Africa and not only in industrialised areas. In addition, high CO and VOC emissions associated with biomass burning result in high $O_3$ production rates as indicated above.

Form the results and discussion presented above it is evident that reducing CO (as well as

anthropogenic and biomass burning VOCs) would be the most efficient control strategy to reduce peak $O_3$ concentrations in the interior of South Africa. It is also imperative to consider the seasonal variation in the CO source strength in managing $O_3$ pollution in continental southern Africa. Anthropogenic emissions of CO, which include emissions from increased household combustion for space heating and cooking during winter, as well as other low-level sources

contributing to increased CO levels associated with the concentration of pollutants during winter such as vehicular emissions, should be targeted to reduce CO emissions. It was also indicated in this study that open biomass burning is a significant source of $O_3$ precursors and it should therefore also be aimed to reduce the influence of regional biomass burning, which is a major source of CO (and VOC) emissions during late winter and early spring, on increased $O_3$

concentrations. However, since open biomass burning in southern Africa is of anthropogenic and natural origin, while $O_3$ concentrations in continental South Africa is also influenced by the trans boundary transport of $O_3$ precursors from open biomass burning occurring in other countries in southern Africa (as indicated above), it is more difficult to control. Nevertheless, open biomass burning caused by anthropogenic practices (e.g. crop residue, pasture

maintenance fires, opening burning of garbage) can be addressed.

## 4. Conclusions

In this study continuous $O_3$ measurements were presented for four sites in the north-eastern

interior in South Africa. Two of these sites, i.e. Welgegund and Botsalano are considered to be regional background sites, while the other two sites were in close proximity to anthropogenic sources, i.e. Marikana and Elandsfontein. Contextualisation of these four sites with other sites in the north-eastern interior of South Africa, indicated that the sites in this study are representative of continental South African $O_3$ levels, while the regional problem of $O_3$ was also shown with $O_3$



concentrations being higher than 40 ppb at many of these sites. The regional problem of $O_3$ in continental South Africa was also signified by the regular exceedance of the 61 ppbv 8-hour moving average South African air quality standard limit at the four sites in this study. $O_3$ levels were generally lower at other background sites in the Southern Hemisphere compared to the

South African sites, while similar seasonal patterns were observed. The seasonal and diurnal $O_3$ patterns observed at the four sites in this study resembled typical trends for $O_3$ in continental South Africa, i.e. $O_3$ concentration peaking in late winter and early spring (cf. Zunckel et al., 2004), and daytime $O_3$ peaks associated with increased photochemical production.

The seasonal $O_3$ trends observed in continental South Africa can mainly be attributed to the seasonal changes in emissions of $O_3$ precursor species, as well as changes in local meteorological conditions and synoptic scale circulation. Increased $O_3$ concentrations in winter at Welgegund, Marikana and Elandsfontein were indicative of increased emissions of $O_3$ precursors from household combustion for space heating and the concentration of low-level

pollutants near the surface. Furthermore, seasonal and diurnal patterns of $O_3$ concentrations at Elandsfontein also reflected high-stack emissions associated with industries in the Mpumalanga Highveld, while temporal $O_3$ patterns at Marikana were indicative of low-level emissions of $O_3$ precursors. A spring maximum was observed at all the sites, which was attributed to increased regional biomass burning during this time. Source area maps of $O_3$ indicated significantly higher

$O_3$ concentrations associated with air masses passing over a region where a large number of open biomass burning occurred, i.e. southern and central Mozambique, southern Zimbabwe and south-eastern Botswana, while CO concentrations were also considerably higher in air masses passing over this region. These source maps indicated that the regional transport of CO associated with biomass burning occurring from June to September in southern Africa is a

significant source of surface $O_3$ in continental South Africa. Furthermore, a very small contribution from the stratospheric intrusion of $O_3$-rich air to surface $O_3$ levels measured at the four sites was indicated.

In the absence of VOC data, the relationship between $O_3$, $NO_x$ and CO at Welgegund,

Botsalano and Marikana was investigated, which indicated a strong dependence of $O_3$ on CO, while $O_3$ levels remained relatively constant (or decrease) with increasing $NO_x$. Although $NO_x$ and VOCs are usually considered as the main precursors in ground-level $O_3$ formation, CO acts together with $NO_x$ and VOCs in the presence of sunlight to drive photochemical $O_3$ formation. In addition, the seasonal changes in the relationship between $O_3$ and precursors species also


reflected the seasonal changes in sources of precursors, i.e. higher CO emissions associated with increased household combustion in winter, and open biomass burning in late winter and spring. The calculation of the $P(O_3)$ from a two year VOC dataset at Welgegund, indicated that at least 40% of $O_3$ production occurred in the VOC-limited regime. These relationships between

$O_3$ concentrations and $P(O_3)$ with $O_3$ precursor species indicated that large parts of the regional background in continental South Africa can be considered CO-limited or VOC-limited, which can be attributed to high anthropogenic emissions of $NO_x$ in the interior of South Africa. It is, however, recommend that future studies investigate more detailed relationships between $NO_x$, CO, VOCs and $O_3$, while concurrent measurement of atmospheric $^\bullet OH$ would also increase the

understanding of surface $O_3$ in this region.

In this paper some new aspects on $O_3$ for the continental South African have been indicated, which must be taken in consideration when $O_3$ mitigation strategies are deployed. These results help to identify the key sources and precursor species for $O_3$ formation, which also highlight the

regional problem of $O_3$ pollution in southern Africa with notably high rural $O_3$ concentrations in areas far removed from pollution sources. It was indicated that CO emissions associated with household combustion and regional open biomass burning should be targeted to reduce $O_3$ concentrations in the interior of South Africa. However, open biomass burning can also be of natural origin, while the influence of regional transport on $O_3$ precursors in continental South

Africa was also evident. In general, the influence of long-range transport must be considered when designing $O_3$ control strategies. A contributing factor to $O_3$ exceedances observed in the north-eastern interior in South Africa is the concentrated location of industries in this area, e.g. nine coal-fired power stations and a petrochemical plant located in this region. Emissions of $O_3$ precursor species could therefore be regulated in this region by enforcing mitigations strategies

on these industries. However, emissions of $O_3$ precursor species related to factors such as regional biomass burning, as well as household combustion associated with poor socio-economic circumstances, provides a bigger challenge for regulators.

**Acknowledgements**

The financial assistance of the National Research Foundation (NRF) towards this research is hereby acknowledged. Opinions expressed and conclusions arrived at are those of the authors and are not necessarily to be attributed to the NRF. We thank the Tropospheric Ozone Assessment Report (TOAR) initiative for providing the surface ozone data used in this





publication. The authors are also grateful to Eskom for supplying the Elandsfontein data. Thanks are also due to Dirk Cillers from NWU for the GIS assistance. V. Vakkari is beneficiary of an AXA Research Fund postdoctoral grant. This work was partly funded by the Academy of Finland Centre of Excellence program (grant no. 272041).

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

50



**Appendix A**

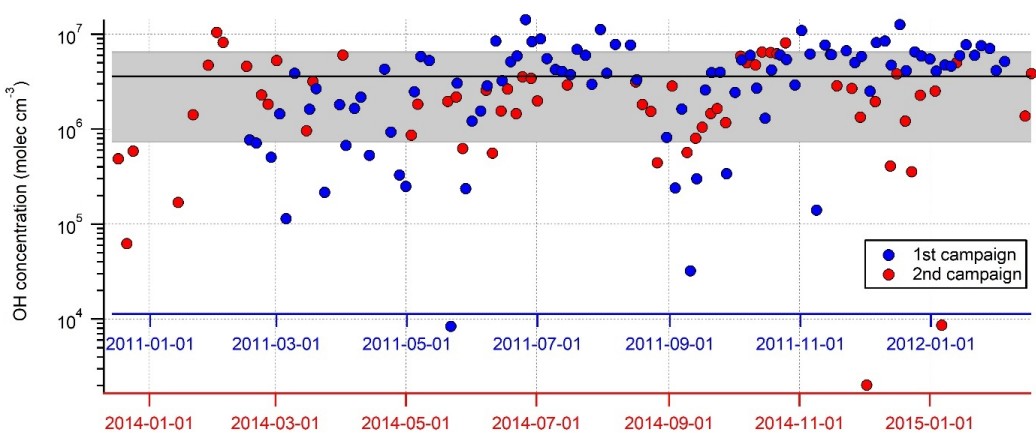

**Fig. A1.** Estimated concentrations of OH radicals using the VOC and $NO_x$ measurements. Production of $HO_x$ was estimated using 41 ppbv of ozone and 42 % relative humidity (both campaign averages for 11:00 LT data) as well as $J(O_3) = 3 \times 10^{-5}$ s$^{-1}$. The black line represents the campaign daytime average, 3.60 ($\pm$ 2.86) $\times 10^6$ molec/cm$^3$.

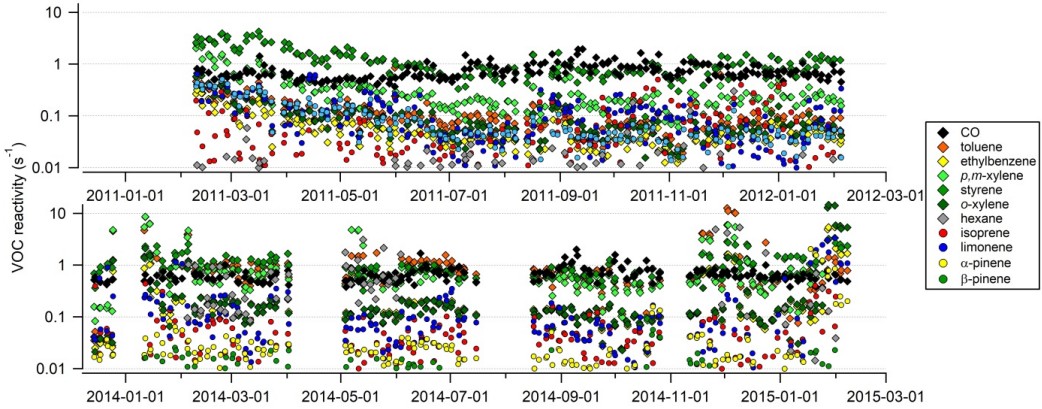

**Fig. A2.** Individual VOC reactivity time series. In the calculation of instantaneous $O_3$ production ($P(O_3)$), CO was treated as a VOC.





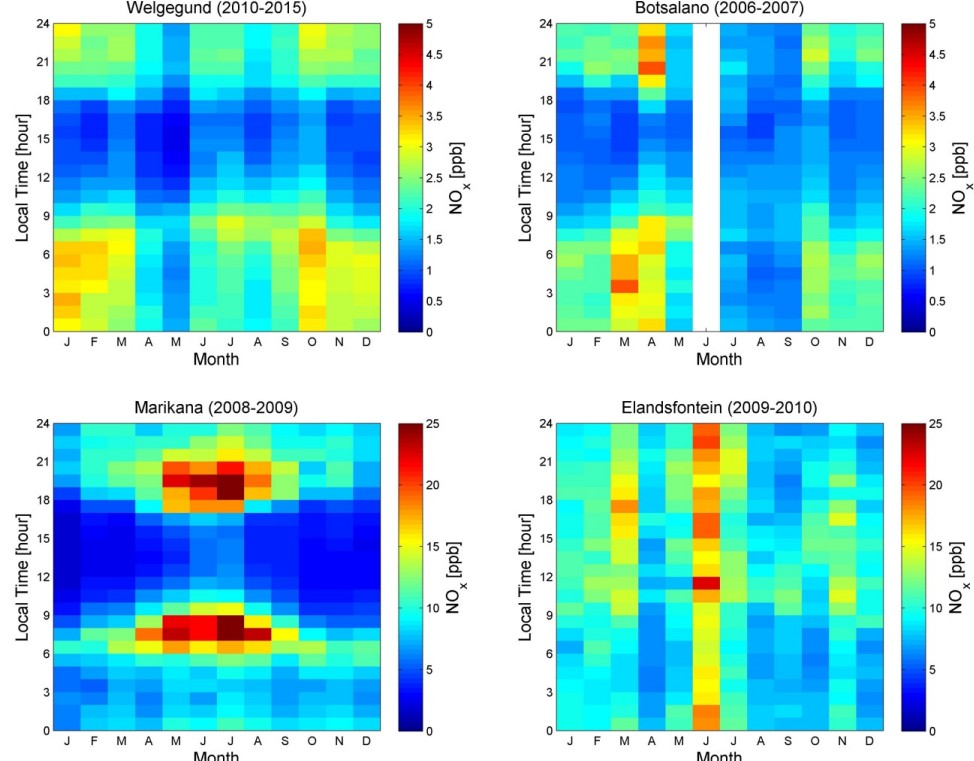

**Fig. A3.**   Seasonal and diurnal variation of $NO_x$ at Welgegund, Botsalano, Marikana and Elandsfontein (median values of $NO_x$ concentration were used).





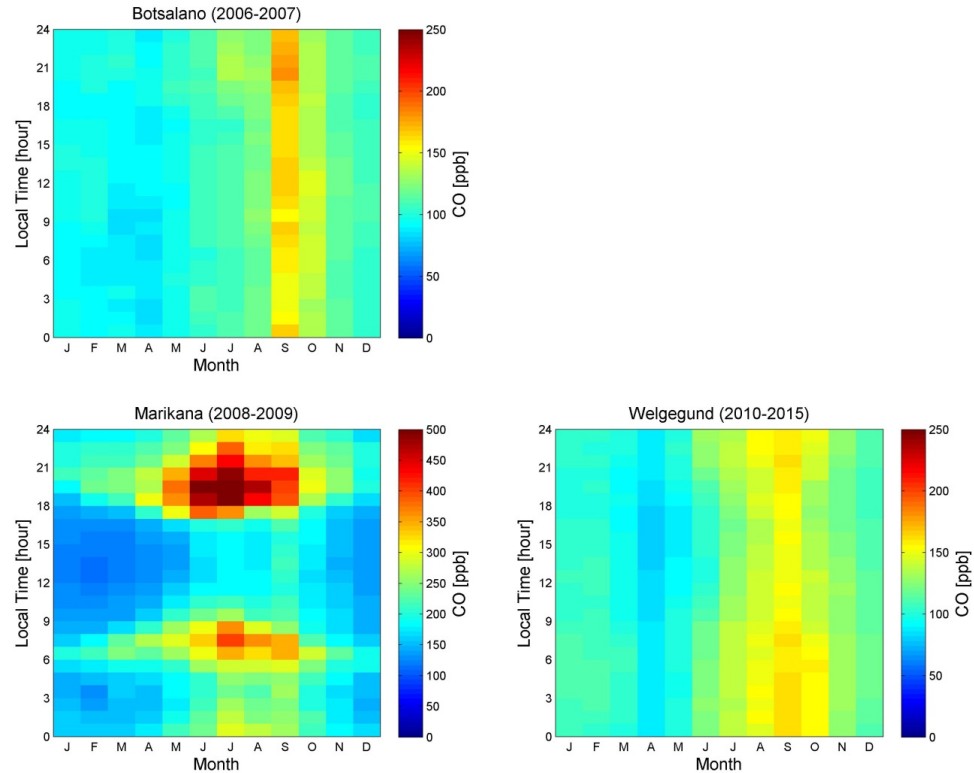

**Fig. A4.**  Seasonal and diurnal variation of CO at Welgegund, Botsalano and Marikana
(median values of CO concentration were used). Note that CO was not measured at
Elandsfontein.





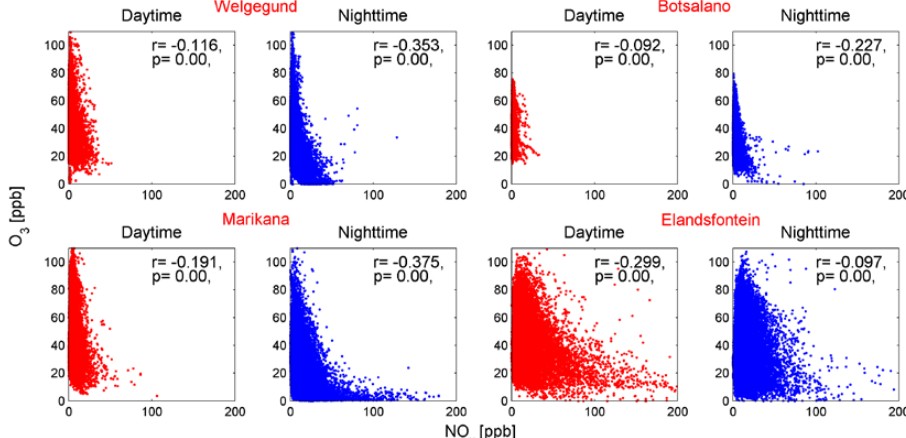

**Fig. A5.** Scatter plots of $O_3$ vs. $NO_x$ for daytime (9:00 a.m. to 4:52 p.m.), and nighttime (5:00 p.m. to 8:52 a.m.) at Welgegund, Botsalano and Marikana and Elandsfontein. The correlation coefficient (r) has a significance level of $p < 10^{-10}$ which means that r is statistically significant ($p < 0.01$).