# Peer review of "Seasonal influences on surface ozone variability in continental South Africa and implications for air quality"

_Atmospheric Chemistry and Physics, 2017_

## Referee Comment (RC1) · Anonymous Referee #1 · 4 Feb 2018

The paper describes observations of ozone and other relevant gas phase species made over a number of years in central South Africa and attempts to evaluate the ozone production regime. The paper describes the sites used and the methodology for assessment.

There are few reported observations of air pollution in Africa and this paper provides a useful description of the regional background conditions present in South Africa. In general, the publication is suitable for publication. However, I have a couple of suggestions which might improve the usefulness of the publication. I outline my major comments and then identify some minor issues.

[Figure]

Source of the ozone. The authors are trying to make the case that CO plays a significant or dominant role in the production of O3 over south Africa - 'Abstract: It was indicated that the appropriate emission control strategy should be CO (and VOC) reduction associated with household combustion and regional open biomass burning to effectively reduce O3 pollution in continental South Africa.' They do this through Figure 6 which shows the that trajectories arriving with high CO are the same trajectories with high O3. They back this up with the arguments from their calculation of ozone production which is essentially ranks the local O3 production for the different VOCs / CO by their OH reactivity. The difficulty with the trajectories argument is that it provides evidence of a common source (biomass burning) but doesn't necessarily show that the CO is leading to the ozone. Biomass burning is known to emit significant quantities of VOCs and NOx into an airmass, which overall leads to O3 production. Attributing the ozone production to the CO specifically from the biomass burning is difficult and probably requires a more detailed analysis than that provided here. Similarly, the local reactivity calculation shows that CO is a significant player in the reactivity, but there are not many datapoints in Figure A2 where the CO is the dominant source of the reactivity. For much of the time it appears that the aromatics, presumably from local industrial activities would outweigh the CO.

Given this, I think that the strength of the comments about the role of CO should probably be toned down. CO is obviously playing a large role here and this is surprising as CO is generally not seen as really leading to regional O3 production. However, I think a policy of reducing both the CO and the VOCs together is likely the story here rather than an emphasis on CO alone. This would probably change the emphasis of sources from those for CO alone (domestic burning, biomass burning) to include some industrial component which would presumably be the source of the aromatic compounds.

Observed concentrations. It would be useful to provide a basic time series of concentration for the key compounds measured at the 4 sites (O3, CO, NO, NO2 etc). The summary plots (Figure 3 and Figure 4) are fine in themselves but it would be useful to

see the full dataset as this would show the scope of the observations and boost the confidence in the quality of the dataset and the subsequent analysis.

The abstract says that much of region is above 40 ppbv of ozone, whereas the corresponding text (Page 13 lines 15) says that this is the case only in the spring time. Figure 3 would suggest that the observational sites are rarely above 40 ppbv. Can this all be clarified? The color scale on Figure 2 makes it almost impossible to define the color for 40 ppbv. Could this be improved and the color scale on Figure 2 lengthened so that the relationship between colors and concentrations is easier to understand?

Reactivity calculation The reactivity calculation is based on the measured CO and the measured VOCs. It is therefore a lower limit. This should be more explicitly explained. There are some obvious missing compounds in this calculation methane, alkanes, alkenes etc. Their concentrations could be estimated. Would they change the perspective offered on whether the site is VOC or NOx limited? Presumably not and it it would only have a slight tendency to move the data-points in figure 11 upwards but not very much? It would be better to make some comments about this head on rather than ignoring it.

Minor comments. The abstract is rather long. Could this be shortened?

The explanation of ozone production at the top of page 3 is a little confused. It starts of saying that the only way to produce ozone is through NO2 photolysis but then says that this doesn't make ozone. Can this be re-phrased to be clearer?

Page 4 line 30. It would be useful to explain what the South Africa AQ standard for O3 is here. It is mentioned in a couple of places in the text but it take us a bit of time to find out what these values are.

The resolution of Figure 7 is rather low. The country names are not clear at the output resolution.

The text at the start of section 3.4 is a little confused. The first sentence says that there

is an absence of VOC data. The next sentence talks about a two-year dataset. It's not obvious what the first sentence therefore means.

I'm not sure that Figure A1 is necessary. Its comes out of the calculation but its isn't really needed for the calculation of P(O3) which essentially just uses the reactivity. Just stating the campaign average value and the variability is enough to show that the calculation is giving a reasonable number.
* * *

---

## Referee Comment (RC2) · Anonymous Referee #2 · 23 Feb 2018

This paper reports four sets of surface ozone measurements in South Africa to explore the spatio-temporal variations as well as the major processes affecting surface ozone variability. Although the measurement data are quite valuable and can enrich the global tropospheric ozone observation database, the current manuscript cannot merit for publication at a high quality journal like ACP. The authors are encouraged to revise the manuscript and submit to another localized journal. I have the following concerns and comments for the author's reference.

Major Concerns:

On the significance of this study: the current manuscript looks more like a report other

than an academic paper. Almost all the results and findings regarding the ozone variations and processes are already well known, except for that the data are newly acquired from South Africa (actually some of the data had been reported in previous studies). The authors need shorten the general description and interpretation of the results and elaborate more about the new findings and significance of the present study.

On the writing of the paper: although the organization and writing of the paper is overall fair, the manuscript is too long and contains a lot of very basic information which I presume the readership of the journal has already known. Some discussions are redundant with each other. For example, the abstract and conclusions are very long and should be largely shortened. The second paragraph in the Introduction (Page 3) describes the ozone formation principles which are very familiar with the community. Seasonal variations of ozone were discussed in Sections 3.1.2 (Fig. 3), 3.2 (Fig. 4), and 3.3 (Fig. 5). The authors are encouraged to remove/shorten such general description and focus on the main findings, and write the paper more concisely.

On the calculation of the ozone production rate: the authors should carefully evaluate if this empirical method is applicable to the environmental conditions in the present study. From the equation in the paper, the P(O3) was calculated as the double reaction rates of VOCs with OH. This assumption may only work to some degree for the high NOx and low VOC conditions. And even under such conditions, the ozone production rate might be also largely underestimated as the contributions of the VOC oxidation products to ozone formation are ignored. Furthermore, the empirical calculation of OH concentrations should be also only applicable to rural atmospheres where ozone photolysis is the dominant OH source, and may be subject to large uncertainty in polluted areas where other radical sources such as HONO and OVOCs photolysis become more important. Therefore, the calculation of P(O3) in this study may be subject to large uncertainty that the authors have to address.

On the "CO-limited ozone formation regime": the authors concluded from the O3-NOx-CO relationship analysis that CO played a significant role in O3 formation in South

Africa (or the so-called "CO-limited O3 formation regime"). I highly suspect that this should be not true. In general, CO is less important than VOCs for ozone formation even though it contributes to a significant fraction of OH reactivity. This is because that the contributions of VOCs can be magnified by not only the ROx radical cycle but also the further reactions of their oxidation intermediates and products. The authors are strongly encouraged to utilize the available data of VOCs, NOx, CO and O3 to perform a photochemical modeling analysis to examine the detailed O3 formation regimes.

Other comments:

Section 2.2: it would be better to provide the detection limit and measurement accuracy of the individual measurements. The traditional NO2 measurements may be subject to positive interference from the catalytic conversion, especially in rural and remote areas. The authors need elaborate more about their NOx measurements.

Figure 2: it would be better to highlight the four measurement sites in the present study in the map, and indicate the prevailing wind directions.

Page 13, Line 1: "Marikana" is a typo?

Section 3.1.2 and Fig. 3: it would be much helpful if the measurement results in East Asia can be also compared to obtain a wider spatial coverage.

Section 3.2: on the interpretation of the late winter and early spring ozone maximum, what are the meteorological conditions (e.g. temperature, solar radiation, etc.) during this period?

Page 19, Lines 1-15: the authors attributed the lower ozone concentrations at Elandsfontein to the high-stack emissions. However, the surface ozone in the industrialized areas can be also titrated by the freshly emitted NOx. It would be helpful for the authors to examine the Ox (Ox=O3+NO2) levels to exclude the effect of NO titration.

Page 28, Lines 11-13: from Fig. 11, most the data points fall in the NOx-limited regime zone. This doesn't support the statement that large part of the regional background of

continental South Africa can be considered VOC-limited.

---

## Author Comment (AC1) · 11 May 2018

The paper describes observations of ozone and other relevant gas phase species made over a number of years in central South Africa and attempts to evaluate the ozone production regime. The paper describes the sites used and the methodology for assessment.

There are few reported observations of air pollution in Africa and this paper provides a useful description of the regional background conditions present in South Africa. In general, the publication is suitable for publication. However, I have a couple of suggestions which might improve the usefulness of the publication. I outline my major comments and then identify some minor issues.

We would like to thank Referee #1 for the positive review of this paper through recognition of the usefulness of this manuscript and deeming this work suitable for publication in ACP. We would also like to thank Referee #1 for the major and minor suggestions made, which were each carefully considered and addressed/implemented in the paper. Below is a point-by-point response to each of these comments/questions. In addition, a marked-up version of the revised manuscript is also provided indicating all changes made throughout the manuscript. The paper was also proofread by a professional language editor.

Source of the ozone. The authors are trying to make the case that CO plays a significant or dominant role in the production of O3 over south Africa - 'Abstract: It was indicated that the appropriate emission control strategy should be CO (and VOC) reduction associated with household combustion and regional open biomass burning to effectively reduce O3 pollution in continental South Africa.' They do this through Figure 6 which shows the that trajectories arriving with high CO are the same trajectories with high O3. They back this up with the arguments from their calculation of ozone production which is essentially ranks the local O3 production for the different VOCs / CO by their OH reactivity. The difficulty with the trajectories argument is that it provides evidence of a common source (biomass burning) but doesn't necessarily show that the CO is leading to the ozone. Biomass burning is known to emit significant quantities of VOCs and NOx into an airmass, which overall leads to O3 production. Attributing the ozone production to the CO specifically from the biomass burning is difficult and probably requires a more detailed analysis than that provided here. Similarly, the local reactivity calculation shows that CO is a significant player in the reactivity, but there are not many datapoints in

Figure A2 where the CO is the dominant source of the reactivity. For much of the time it appears that the aromatics, presumably from local industrial activities would outweigh the CO.

Given this, I think that the strength of the comments about the role of CO should probably be toned down. CO is obviously playing a large role here and this is surprising as CO is generally not seen as really leading to regional O3 production. However, I think a policy of reducing both the CO and the VOCs together is likely the story here rather than an emphasis on CO alone. This would probably change the emphasis of sources from those for CO alone (domestic burning, biomass burning) to include some industrial component which would presumably be the source of the aromatic compounds.

We agree with Referee #1 that the strength of the role of CO on $O_3$ formation should be toned down and that policy should focus on reducing both CO and VOCs. However, the significance of CO to $O_3$ formation for this region where biogenic VOCs are relatively less abundant (Jaars et al., 2016) is indicated in this paper, especially through the correlation plots in Fig. 9 and 10 (now Fig. 8 and 9 in the revised manuscript). Therefore the strength of the role of CO on $O_3$ was toned down in the manuscript in different sections and the contribution of VOCs indicated, without compromising the significance of CO on $O_3$ formation shown for this region in this paper. Section where changes were made are:

Abstract:

[revised manuscript text omitted]

Observed concentrations. It would be useful to provide a basic time series of concentration for the key compounds measured at the 4 sites (O3, CO, NO, NO2 etc). The summary plots (Figure 3 and Figure 4) are fine in themselves but it would be useful to see the full dataset as this would show the scope of the observations and boost the confidence in the quality of the dataset and the subsequent analysis.

Basic time series of $O_3$, $NO_x$ and CO were included in the Appendix in Fig. A2, A7 and A8, respectively. The following was also included in Section 3.2 (now Section 3.1) and Section 3.3.1 (now Section 3.4.1) referencing these time series plots:

"In Fig. 2 the monthly and diurnal variation for $O_3$ concentrations measured at the four sites in this study are presented (time series plotted in Fig. A2). Although there is some variability between the sites, monthly…"

"…open biomass burning emissions (i.e. $NO_x$ and CO indicated in Fig. A3 and Fig. A4, respectively – time series plotted in Fig. A7 and A8), while $O_3$ levels at Botsalano were predominantly…"

The abstract says that much of region is above 40 ppbv of ozone, whereas the corresponding text (Page 13 lines 15) says that this is the case only in the spring time. Figure 3 would suggest that the observational sites are rarely above 40 ppbv. Can this all be clarified? The color scale on Figure 2 makes it almost impossible to define the color for 40 ppbv. Could this be improved and the color scale on Figure 2 lengthened so that the relationship between colors and concentrations is easier to understand?

We agree with Referee #1 on both aspects indicated here. Although $O_3$ concentrations exceeded 40 ppb on a daily basis at most of the sites throughout the year as indicated in Fig. 4 (now Fig. 2 in the revised manuscript), this is not clearly indicated in Fig. 3 (now Fig. 4 in the revised manuscript), since mean $O_3$ concentrations are presented in this figure. The spatial map, i.e. Fig. 2 (now Fig. 3) compiled from average spring $O_3$ concentrations do, however, indicate relatively high $O_3$ levels across the region, albeit not necessarily above 40 ppb. Therefore the text referring to the regional $O_3$ problem in the Abstract, Section 3.1.1 (now Section 3.2.1) and the Conclusions was changed as follows:

Abstract

"…four sites in continental South Africa was conducted. The regional $O_3$ problem was evident with $O_3$ concentrations regularly exceeding the South African air quality standard limit, while $O_3$ levels were higher compared to other background sites in the Southern Hemisphere. The temporal $O_3$ patterns observed at the four sites…"

Section 3.1.1 (now Section 3.2.1)

"…Johannesburg-Pretoria megacity, while the rural Vaalwater site in the north also has significantly higher $O_3$ levels. From Fig. 3 it is evident that $O_3$ can be considered a regional problem with $O_3$ concentrations being relatively high across continental South Africa during spring. Fig. 3 also clearly indicates that the four research sites…"

Conclusions

"A spatial distribution map of $O_3$ levels in the interior of South Africa indicated the regional $O_3$ problem in continental South Africa, which was signified by the regular exceedance of the South African air quality standard limit. The seasonal and diurnal $O_3$ patterns observed at the four sites in this study resembled typical trends for $O_3$ in continental…"

The colour scale in Fig. 2 (now Fig. 3) was also improved by lengthening the scale and adding more values on the scale.

Reactivity calculation The reactivity calculation is based on the measured CO and the measured VOCs. It is therefore a lower limit. This should be more explicitly explained. There are some obvious missing compounds in this calculation methane, alkanes, alkenes etc. Their concentrations could be estimated. Would they change the perspective offered on whether the site is VOC or NOx limited? Presumably not and it it would only have a slight tendency to move the data-points in figure 11 upwards but not very much? It would be better to make some comments about this head on rather than ignoring it.

We agree with Referee #1 that our VOC reactivity estimates are a lower limit. We can only speciate a fraction of the VOCs present in our grab samples. Although we are likely measuring the major contributors to VOC reactivity at Welgegund such as *o*-xylene, CO, styrene, *p,m*-xylene, toluene, ethylbenzene limonene, isoprene, α-pinene, β-pinene, hexane (depicted in Fig. A2), we are certainly missing methane that could also contribute to increasing the VOC reactivity. Yet, assuming a global ambient concentrations of 1.85 ppm and a rate of oxidation by OH radicals of $6.68 \times 10^{-15}$ cm$^3$ molec$^{-1}$ s$^{-1}$ (Srinivasan et al., 2005) would lead to a VOC reactivity of 0.3 s$^{-1}$. Thus, as Referee #1 mentions, a slight tendency to move the data points upwards by 0.3 s$^{-1}$. However, this shift would not impact the O$_3$ production regime inferred. Nonetheless, our VOC dataset is quite comprehensive and includes 6 trace gases, 19 biogenic VOCs and 20 anthropogenic VOCs, including 13 aromatic and 7 aliphatic compounds as presented in Jaars, et al. 2014, 2016.

To further address Referee #1 comments, we have also amended our tone in the paragraph to discuss all sources of error and estimation in detail in an attempt to be transparent with our calculations and assumptions. Consequently, the paragraph presenting the model in Section 2.4 has been entirely rewritten to address these issues as follows:

[revised manuscript text omitted]

In addition Section 3.4.3 (now Section 3.5.3) was also rewritten to indicate limitations of the model as follows:

"In Fig. 10 $P(O_3)$ as a function of VOC reactivity calculated from the available VOC dataset for Welgegund (Section 2.4) and $NO_2$ concentrations is presented. $O_3$ production at Welgegund during two field campaigns, specifically at 11:00 LT, were found to range between 0 and 10 ppbv $h^{-1}$. The average $P(O_3)$ over the 2011 to 2012 and the 2014 to 2015 campaigns combined were $3.0 \pm 1.9$ ppbv $h^{-1}$ and $3.2 \pm 3.0$ ppbv $h^{-1}$, respectively. The dashed black line in Fig. 10, called the ridge line, separates the $NO_x$- and VOC-limited regimes. To the left of the ridge line is the $NO_x$-limited regime, when $O_3$ production increases with increasing $NO_x$ concentrations. The VOC-limited regime is to the right of the ridge line, when $O_3$ production decreases with increasing $NO_x$. According to the $O_3$ production plot presented, approximately 40% of the data is found in the VOC-limited regime area, which would support the regional $O_3$ analysis conducted for continental South Africa in this study. However, the $O_3$ production plot for Welgegund transitions between $NO_x$- and VOC-limited regimes with Welgegund being in a $NO_x$-limited production regime the majority of the time, especially when $NO_x$ concentrations are very low (<1 ppb). As indicated in Section 2.4, limitations to this analysis include limited VOC speciation data, as well as a single time-of-day grab sample. The $O_3$ production rates can therefore only be inferred at 11:00 am LT despite $O_3$ concentrations peaking during the afternoon at Welgegund. Therefore, clean background air $O_3$ production is most-likely $NO_x$-limited (Tiitta et al., 2014), while large parts of the regional background of continental South Africa can be considered VOC-limited."

**Minor comments.**

The abstract is rather long. Could this be shortened?

We agree with Referee #1 (and Referee #2) that the abstract is too long, which was significantly shortened in the revised manuscript as follows:

"Although elevated surface ozone ($O_3$) concentrations are observed in many areas within southern Africa, few studies have investigated the regional atmospheric chemistry and dominant atmospheric processes driving surface $O_3$ formation in this region. Therefore an assessment of comprehensive continuous surface $O_3$ measurements performed at four sites in continental South Africa was conducted. The regional $O_3$ problem was evident with $O_3$ concentrations regularly exceeding the South African air quality standard limit, while $O_3$ levels were higher compared to other background sites in the Southern Hemisphere. The temporal $O_3$ patterns observed at the four sites resembled typical trends for $O_3$ in continental South Africa with $O_3$ concentration peaking in late winter and early spring. Increased $O_3$ concentrations in winter were indicative of increased emissions of $O_3$ precursors from household combustion and other low-level sources, while a spring maximum observed at all the sites was attributed to increased regional biomass burning. Source area maps of $O_3$ and CO indicated significantly higher $O_3$ and CO concentrations associated with air masses passing over a region with increased seasonal open biomass burning, which indicated CO associated with open biomass burning as a major source of $O_3$ in continental South Africa. A strong correlation between $O_3$ on CO was observed, while $O_3$ levels remained relatively constant or decreased with increasing $NO_x$, which supports a VOC-limited regime. The instantaneous production rate of $O_3$ calculated at Welgegund indicated that ~40% of $O_3$ production occurred in the VOC-limited regime. The relationship between $O_3$ and precursor species suggests that continental South Africa can be considered VOC-limited, which can be attributed to high anthropogenic emissions of $NO_x$ in the interior of South Africa. The study indicated that the most effective emission control strategy to reduce $O_3$ levels in continental South Africa should be CO and VOC reduction mainly associated with household combustion and regional open biomass burning."

The explanation of ozone production at the top of page 3 is a little confused. It starts of saying that the only way to produce ozone is through NO2 photolysis but then says that this doesn't make ozone. Can this be re-phrased to be clearer?

$NO_2$ photolysis is the only known way through which $O_3$ is produced in the troposphere. However the resultant $O_3$ reacts with NO to form $NO_2$, which will again undergo photolysis to produce $O_3$ and NO resulting in a null cycle, i.e. the photostationary state (PSS). This equilibrium is disturbed when peroxy radicals alter the PSS producing $NO_2$, which lead to the formation of $O_3$ in excess of the null cycle.

This entire paragraph in the Introduction was changed in accordance with a suggestion made by Referee #2 and to address the confusion indicated by Referee #1 in the above mentioned comment as follows:

"Tropospheric $O_3$ concentrations are regulated by three processes, i.e. chemical production/destruction, atmospheric transport and losses to surface through dry deposition (Monks et al., 2015). The photolysis of nitrogen dioxide ($NO_2$) in the presence of sunlight is the only known way of producing $O_3$ in the troposphere (Logan, 1985). $O_3$ can recombine with nitric oxide (NO) to regenerate $NO_2$, which will again undergo photolysis to regenerate $O_3$ and NO. This continuous process is known as the $NO_x$-dependent photostationary state (PSS) and results in no net production or consumption of ozone (null cycle). However, net production of $O_3$ in the troposphere occurs outside the PSS when peroxy radicals ($HO_2$ and $RO_2$) alter the PSS by oxidising NO to produce 'new' $NO_2$ (Cazorla and Brune, 2010) resulting in net $O_3$ production. The main source of these peroxy radicals in the atmosphere is the reaction of the hydroxyl radical ($OH^\bullet$) with volatile organic compounds (VOCs) or carbon monoxide (CO) (Cazorla and Brune, 2010)."

Page 4 line 30. It would be useful to explain what the South Africa AQ standard for O3 is here. It is mentioned in a couple of places in the text but it take us a bit of time to find out what these values are.

We thank Referee #1 for pointing this out. The following has been included in the Introduction:

"…provincial governments, local municipalities and industries (http://www.saaqis.org.za). High $O_3$ concentrations are observed in many areas within the interior of South Africa that exceed the South African standard $O_3$ limit, i.e. an 8-hour moving average of 61 ppb (e.g. Laakso et al., 2013). These exceedances can be attributed to high anthropogenic…"

The resolution of Figure 7 is rather low. The country names are not clear at the output resolution.

We agree with Referee #1 and have improved the resolution, text size and marker sizes. The modified figures are presented below.

[Figure]

**Fig. 7.**     Spatial distribution of fires in 2007, 2010 and 2015 from MODIS burnt area product. Blue stars indicate (from left to right) Botsalano, Welgegund, Marikana and Elandsfontein.

The text at the start of section 3.4 is a little confused. The first sentence says that there is an absence of VOC data. The next sentence talks about a two-year dataset. It's not obvious what the first sentence therefore means.

We completely agree with the confusion/inconsistence in these two sentences indicated by Referee #1. What was meant here is that no continuous measurement data existed for VOCs for any of the sites. However, there was VOC data available for Welgegund from VOC measurements conducted with adsorbent tubes during two sampling campaigns, which could be used to calculate the instantaneous production rate of $O_3$. Therefore the text was changed as follows to clarify the confusion:

"The relationship between $O_3$, $NO_x$ and CO was used as an indicator to infer the $O_3$ production regime at Welgegund, Botsalano and Marikana (no CO measurements were conducted at Elandsfontein as indicated above), since no continuous VOC measurements were conducted at each of these sites. However, as indicated in Section 2.4, a two-year VOC dataset was available for Welgegund (Jaars et al., 2016; Jaars et al., 2014), which was used to calculate the instantaneous production rate of $O_3$ as a function of $NO_2$ levels and VOC reactivity (Geddes et al., 2009; Murphy et al., 2006)."

I'm not sure that Figure A1 is necessary. Its comes out of the calculation but its isn't really needed for the calculation of P(O3) which essentially just uses the reactivity. Just stating the campaign average value and the variability is enough to show that the calculation is giving a reasonable number.

Figure A1 was removed from the Appendix as suggested by Referee #1. In response to a previous comment by Referee #1, the paragraph presenting the model in Section 2.4 was also entirely rewritten.

---

## Author Comment (AC2) · 11 May 2018

This paper reports four sets of surface ozone measurements in South Africa to explore the spatio-temporal variations as well as the major processes affecting surface ozone variability. Although the measurement data are quite valuable and can enrich the global tropospheric ozone observation database, the current manuscript cannot merit for publication at a high quality journal like ACP. The authors are encouraged to revise the manuscript and submit to another localized journal. I have the following concerns and comments for the author's reference.

We would like to thank Referee #2 for reviewing this manuscript and indicating that valuable measurement data are presented, which can enrich the global tropospheric ozone observation database. However, in view of the positive review of Referee #1 of this paper, who deemed this work suitable for publication in ACP, we do consider that the work presented in this paper does merit publication in ACP. Referee #1 also highlighted that few observations of air pollution in general are reported for Africa and that a useful description of regional background conditions are presented for South Africa. The results presented in this paper are also considered to be novel and relevant for *southern* Africa, and not only South Africa, which indicate that it is not only of local interest. The relevance and significance of atmospheric measurements conducted in South Africa are also emphasised by the solid body of papers published in ACP during the last few years on atmospheric studies conducted in South Africa. Other original aspects in this paper include: 1) the use of the Tropospheric Ozone Assessment Report data to compile a surface plot to indicate ozone spatial variations for this region; and 2) relating the ozone spring peak in this region to CO and VOCs (and co-emitted species) associated with biomass burning rather than biogenic VOCs. None of these aspects have been indicated in previous papers published for this region. Also by addressing each of the major and minor comments made by both referees, the scientific relevance of this paper was further improved.

We would also like to thank Referee #2 for major concerns raised and minor comment made, which were each carefully considered and addressed/implemented in the paper. Below is a point-by-point response to each of these comments/questions. In addition, a marked-up version of the revised manuscript is also provided indicating all changes made throughout the manuscript. The paper was also proofread by a professional language editor.

Major Concerns:

On the significance of this study: the current manuscript looks more like a report other than an academic paper. Almost all the results and findings regarding the ozone variations and processes are already well known, except for that the data are newly acquired from South Africa (actually some of the data had been reported in previous studies). The authors need shorten the general description and interpretation of the results and elaborate more about the new findings and significance of the present study.

The novelty and significance of this study are argued in the response to the general introductory comment made by Referee #2. The manuscript in general was shorten (by ~1300 words as indicated in the marked-up version of the revised manuscript) and written more concisely through addressing comments/suggestions made by Referee #1 and Referee #2.

On the writing of the paper: although the organization and writing of the paper is overall fair, the manuscript is too long and contains a lot of very basic information which I presume the readership of the journal has already known. Some discussions are redundant with each other. For example, the abstract and conclusions are very long and should be largely shortened. The second paragraph in the Introduction (Page 3) describes the ozone formation principles which are very familiar with the community. Seasonal variations of ozone were discussed in Sections 3.1.2 (Fig. 3), 3.2 (Fig. 4), and 3.3 (Fig. 5). The authors are encouraged to remove/shorten such general description and focus on the main findings, and write the paper more concisely.

We agree with Referee #2 that certain sections in the manuscript are too long and therefore these sections were shortened to exclude basic information and repetition. In addition, the Results section was also restructured in order to contribute to a more concise manuscript:

3.1     Temporal variation of $O_3$

3.2     Spatial distribution of $O_3$ in continental South Africa

3.3     Comparison with international sites

3.4     Sources contributing to surface $O_3$ in continental South Africa

3.4.1   Anthropogenic and open biomass burning emissions

3.4.2   Stratospheric $O_3$

3.5     Insights into the $O_3$ production regime

3.5.1   The relationship between $NO_x$, CO and $O_3$

3.5.2   Seasonal change in $O_3$-precursors relationship

3.5.3   $O_3$ production rate

3.6     Implications for air quality management

3.6.1   Ozone exceedances

**3.6.2    $O_3$ control strategies**

Shortened/re-written sections include:

[revised manuscript text omitted]

On the calculation of the ozone production rate: the authors should carefully evaluate if this empirical method is applicable to the environmental conditions in the present study. From the equation in the paper, the P(O3) was calculated as the double reaction rates of VOCs with OH. This assumption may only work to some degree for the high NOx and low VOC conditions. And even under such conditions, the ozone production rate might be also largely underestimated as the contributions of the VOC oxidation products to ozone formation are ignored. Furthermore, the empirical calculation of OH concentrations should be also only applicable to rural atmospheres where ozone photolysis is the dominant OH source, and may be subject to large uncertainty in polluted areas where other radical sources such as HONO and OVOCs photolysis become more important. Therefore, the calculation of P(O3) in this study may be subject to large uncertainty that the authors have to address.

We agree with Referee #2 that the P(O3) model utilised in this study may be subject to uncertainty. However, it can only be our current best tool for estimating P(O3), and we have rewritten the paragraph on ozone production in Section 2.4 to highlight the assumptions made and the caveats of this model as follows:

[revised manuscript text omitted]

On the "CO-limited ozone formation regime": the authors concluded from the O3-NOxCO relationship analysis that CO played a significant role in O3 formation in South Africa (or the so-called "CO-limited O3 formation regime"). I highly suspect that this should be not true. In general, CO is less important than VOCs for ozone formation even though it contributes to a significant fraction of OH reactivity. This is because that the contributions of VOCs can be magnified by not only the ROx radical cycle but also the further reactions of their oxidation intermediates and products. The authors are strongly encouraged to utilize the available data of VOCs, NOx, CO and O3 to perform a photochemical modeling analysis to examine the detailed O3 formation regimes.

This point was also raised by Referee #1 who suggested that the comments on the strength of the role of CO on $O_3$ formation should be toned down and that policy should focus on reducing both CO and VOCs. However, the significance of CO to $O_3$ formation for this region where biogenic VOCs are relatively less abundant (Jaars et al., 2016) is indicated in this paper, especially through the correlation plots in Fig. 9 and 10 (now Fig. 8 and 9 in the revised manuscript). In addition, Referee #1 also recognised the large role of CO on $O_3$ formation shown in this study. In view of the comment of Referee #1, the strength of the role of CO on $O_3$ was toned down in the manuscript in different sections and the contribution of VOCs indicated, without compromising the significance of CO on $O_3$ formation shown for this region in this paper. Section where changes were made are:

Abstract:

[revised manuscript text omitted]

We agree with Referee #2 that photochemical modeling would greatly assist in establishing $O_3$ formation regime. However, comprehensive modeling was beyond the scope of this paper (which is already long as indicated by Referee #2) since this was a measurement study. However, this is an important future recommendation that was included in the Conclusions section:

"It is, however, recommended that future studies investigate more detailed relationships between $NO_x$, CO, VOCs and $O_3$ through photochemical modelling analysis, while concurrent measurement of atmospheric VOCs and $^\bullet OH$ would also contribute to the better understanding of surface $O_3$ in this region."

Other comments:

Section 2.2: it would be better to provide the detection limit and measurement accuracy of the individual measurements. The traditional NO2 measurements may be subject to positive interference from the catalytic conversion, especially in rural and remote areas. The authors need elaborate more about their NOx measurements.

NO$_2$ levels determined with the Teledyne 200AU NO/NO$_x$ analyser (used at three of sites) were compared with NO$_2$ concentrations measured with a quantum cascade laser used for NO$_2$ flux measurements at Welgegund, which indicated very good comparison between these two instruments. Therefore we have a high level of confidence in the NO$_x$ levels measured with the chemiluminescent measurement techniques at the four sites.

In view of our effort to shorten the paper and exclude basic information as suggested by Referee #2, we do not deem it necessary to elaborate more on the NOx measurements. In addition, the NOx measurements were used to as supportive date to assist in interpreting O$_3$ measurements. However, a sentence on the comparison with the QCL instrument could be included if requested by Referee #2.

Figure 2: it would be better to highlight the four measurement sites in the present study in the map, and indicate the prevailing wind directions.

The author agree. We have improved the map, by including smaller overlay back trajectory maps of the 4 four study sites, which indicates the air mass movement patterns towards the afore-mentioned sites. The four measurement sites in the present study were also highlighted. This map is indicated below:

[Figure]

**Fig. 2.** The main (central map) indicating spatial distribution of mean surface O₃ levels during springtime over the north-eastern interior of southern Africa ranging between 23.00 ° S and 29.03 ° S,

and 25.74 ⁰ E and 32.85 ⁰ E. The data for all sites were averaged for years when the ENSO cycle was not present (by examining SST anomalies in the Niño 3.4 region). Black dots indicate the sampling sites. The smaller maps (top and bottom) indicate 96-hour overlay back trajectory maps for the four main study sites, over the corresponding springtime periods.

Page 13, Line 1: "Marikana" is a typo?

We thank Referee #2 for pointing out this typo, which was changed as follows:

"…during springtime (S-O-N), when $O_3$ is usually at a maximum as indicated above. The mean $O_3$ concentration over continental South Africa ranged from 20 ppb to 60 ppb during spring. From Fig. 3 it can be seen that $O_3$ concentrations at the industrial sites Marikana and Elandsfontein were higher than $O_3$ levels at Botsalano and Welgegund. As mentioned previously, Elandsfontein…"

Section 3.1.2 and Fig. 3: it would be much helpful if the measurement results in East Asia can be also compared to obtain a wider spatial coverage.

Section 3.1.2 and Fig. 3 (now Section 3.3 and Fig. 4) were improved in accordance with the above general comment on the writing of the paper. The seasonal patterns of additional South African sites were removed from Fig. 3 (now Fig. 4) and only Welgegund was used in the comparison to other Southern Hemisphere sites since it had the most extensive data record of all the sites reported on in this study. Therefore Fig. 3 (now Fig. 4) was separated into two figures presenting seasonal patterns for Southern- and Northern Hemisphere sites. In addition, two East Asian sites, i.e. Ryori, Japan and Seokmo-Ri Ga, South Korea, which could be obtained from the TOAR database, were included in the comparison.

Section 3.1.2 (now Section 3.3 and Fig. 4):

"In an effort to contextualise the $O_3$ levels measured in this study, the monthly $O_3$ concentrations measured at Welgegund were compared to monthly $O_3$ levels measured at monitoring sites in other parts of the world (downloaded from the JOIN web interface https://join.fz-juelich.de (Schultz et al., 2017)) as indicated in Fig. 4. Welgegund was used in the comparison since it had the most extensive data record, while the measurement time period considered was from May 2010 to December 2014. The seasonal $O_3$ cycles observed at other sites in the Southern Hemisphere are comparable to the seasonal cycle at Welgegund with slight variations in the time of year when $O_3$ peaks as indicated in Fig. 4. Cape Grim, Australia; GoAmazon T3 Manancapuru, Brazil; Ushuaia, Argentina; and El Tololo, Chile are regional background GAW (Global Atmosphere Watch) stations with $O_3$ levels lower than the South African sites. However, the $O_3$ concentrations at El Tololo, Chile are comparable to Welgegund. Oakdale, Australia and Mutdapliiy, Australia are semi-rural and rural locations, which are

influenced by urban and industrial pollution sources, which also had lower $O_3$ concentrations compared to Welgegund.

[Figure]

**Fig. 4.** Seasonal cycle of $O_3$ at rural sites in other parts of the world. The black dot indicate monthly median (50th percentile) and the upper and lower limits the 25th and 75th percentile, respectively for monthly $O_3$ concentrations. The data is averaged from May 2010 to December 2014, except in a few instances where 2014 data was not available.

The Northern Hemispheric $O_3$ peak over mid-latitude regions is similar to seasonal patterns in the Southern Hemisphere where a springtime $O_3$ maximum is observed (i.e. Whiteface Mountain Summit, Beltsville, Ispra, Ryori and Seokmo-Ri Ga). However, there are other sites in the Northern Hemisphere where a summer maximum is more evident (Vingarzan, 2004), i.e. Joshua Tree and Hohenpeissenberg. The discernible difference between the hemispheres is that the spring maximum in the Southern Hemisphere refers to maximum $O_3$ concentrations in late winter and early spring, whilst in the Northern Hemisphere it refers to a late spring and early summer $O_3$ maximum (Cooper et al., 2014). The spring maximum in the Northern Hemisphere is associated with stratospheric intrusions (Zhang et al., 2014; Parrish et al., 2013), while the summer maximum is associated with photochemical $O_3$ production from anthropogenic emissions of $O_3$ precursors being at its highest (Logan, 1985; Chevalier et al., 2007). Maximum $O_3$ concentrations at background sites in the United States and Europe are similar to values at Welgegund in spring with the exception of Joshua Tree National Park in the United States that had significant higher $O_3$ levels. This is most-likely due its high elevation and deep boundary layer (~4 km asl) during spring and summer allowing free tropospheric $O_3$ to be more effectively mixed down to the surface (Cooper et al., 2014). Maximum $O_3$ levels at the two sites in East Asia (Ryori and Seokmo-Ri Ga) were also generally higher than at Welgegund, especially at Seokmo-Ri Ga."

Section 3.2: on the interpretation of the late winter and early spring ozone maximum, what are the meteorological conditions (e.g. temperature, solar radiation, etc.) during this period?

We have included monthly averages of temperature, relative humidity, global radiation and wind speed, as well as total monthly rainfall, for the $O_3$ measurement period at Welgegund in the Appendix (Fig. A3) to indicate typical meteorological conditions for this part of South Africa. The following sentence was also added to Section 3.2 (now Section 3.1)

"In Fig. A3 monthly averages of meteorological parameters and total monthly rainfall for Welgegund are presented to indicate typical seasonal meteorological patterns for continental South Africa."

Page 19, Lines 1-15: the authors attributed the lower ozone concentrations at Elandsfontein to the high-stack emissions. However, the surface ozone in the industrialized areas can be also titrated by the freshly emitted NOx. It would be helpful for the authors to examine the Ox (Ox=O3+NO2) levels to exclude the effect of NO titration.

We agree with Referee #2 that low-level freshly emitted $NO_x$ could also contribute to $O_3$ titration at industrial sites. This was clearly shown for Marikana, where $NO_x$ emissions were predominantly associated with low-level emissions as indicated by the bimodal diurnal $NO_x$ peaks (Fig. A3, now Fig. A4), as well as the nighttime titration of $O_3$ (Fig. 4, now Fig. 2) as discussed in this section. However, the diurnal $NO_x$ pattern at Elandsfontein clearly shows daytime $NO_x$ peaks (Fig. A3, now Fig. A4) resulting from downward mixing of high-stack $NO_x$ emissions as the predominant source of $NO_x$ at Elandsfontein, while the nighttime titration of $O_3$ is also less significant compared to Marikana (Fig. 4, now Fig. 2). This was also indicated in by Collett et al., 2010 for Elandsfontein. Therefore, in our view the major contributing sources to $O_3$ titrations at these two industrial sites are adequately indicated by the $O_3$ (Fig. 4, now Fig. 2) and $NO_x$ (Fig. A3, now Fig. A4) temporal patterns presented in the manuscript.

Page 28, Lines 11-13: from Fig. 11, most the data points fall in the NOx-limited regime zone. This doesn't support the statement that large part of the regional background of continental South Africa can be considered VOC-limited.

Most of the points on the graph do indeed lie in the NOx-limited regime. Since Welgegund is the only station for which VOCs have been measured in South Africa, we cannot directly compare $O_3$ production with other sites. It would be likely that $O_3$ production at Welgegund may not be representative of regional $O_3$ production conditions across the country. However, ~ 40% of the data is found in the VOC-limited regime area, which would support the regional $O_3$ analysis conducted for continental South Africa in this study. In addition, $O_3$ production is mainly in the NOx limited regime at low NOx concentrations. Therefore, clean background air $O_3$ production is most-likely $NO_x$-limited, while large parts of the regional background of continental South Africa can be considered VOC-limited. We also feel that this comparison truly highlights the need for further research to reconcile $O_3$ production at

Welgegund and the positive correlations observed between $O_3$ and CO at most of our other sites that would indicate a rather VOC-limited regime. Therefore Section 3.4.3 (now Section 3.5.3) on $O_3$ production rate was rewritten as follows:

"In Fig. 10 $P(O_3)$ as a function of VOC reactivity calculated from the available VOC dataset for Welgegund (Section 2.4) and $NO_2$ concentrations is presented. $O_3$ production at Welgegund during two field campaigns, specifically at 11:00 LT, were found to range between 0 and 10 ppbv $h^{-1}$. The average $P(O_3)$ over the 2011 to 2012 and the 2014 to 2015 campaigns combined were $3.0 \pm 1.9$ ppbv $h^{-1}$ and $3.2 \pm 3.0$ ppbv $h^{-1}$, respectively. The dashed black line in Fig. 10, called the ridge line, separates the $NO_x$- and VOC-limited regimes. To the left of the ridge line is the $NO_x$-limited regime, when $O_3$ production increases with increasing $NO_x$ concentrations. The VOC-limited regime is to the right of the ridge line, when $O_3$ production decreases with increasing $NO_x$. According to the $O_3$ production plot presented, approximately 40% of the data is found in the VOC-limited regime area, which would support the regional $O_3$ analysis conducted for continental South Africa in this study. However, the $O_3$ production plot for Welgegund transitions between $NO_x$- and VOC-limited regimes with Welgegund being in a $NO_x$-limited production regime the majority of the time, especially when $NO_x$ concentrations are very low (<1 ppb). As indicated in Section 2.4, limitations to this analysis include limited VOC speciation data, as well as a single time-of-day grab sample. The $O_3$ production rates can therefore only be inferred at 11:00 am LT despite $O_3$ concentrations peaking during the afternoon at Welgegund. Therefore, clean background air $O_3$ production is most-likely $NO_x$-limited (Tiitta et al., 2014), while large parts of the regional background of continental South Africa can be considered VOC-limited."